# Mesenchymal–Epithelial Transition in Fibroblasts of Human Normal Lungs and Interstitial Lung Diseases

**DOI:** 10.3390/biom11030378

**Published:** 2021-03-04

**Authors:** Carina Becerril, Martha Montaño, José Cisneros, Criselda Mendoza-Milla, Annie Pardo, Blanca Ortiz-Quintero, Moisés Selman, Carlos Ramos

**Affiliations:** 1Cellular Biology Laboratory, Department of Research in Pulmonary Fibrosis, Instituto Nacional de Enfermedades Respiratorias Ismael Cosío Villegas (INER), Mexico City 14080, Mexico; lcbb6@hotmail.com (C.B.); mamora572002@yahoo.com.mx (M.M.); criselda.mendoza@gmail.com (C.M.-M.); 2Laboratory of Pulmonary Biopathology INER—Ciencias UNAM, Department of Research in Pulmonary Fibrosis, Instituto Nacional de Enfermedades Respiratorias Ismael Cosío Villegas (INER), Mexico City 14080, Mexico; jcisneros828@gmail.com; 3Laboratory of Pulmonary Biopathology, Facultad de Ciencias, Universidad Nacional Autónoma de Mexico (UNAM), Ciudad de México 14080, Mexico; annie.pardo@ciencias.unam.mx; 4Investigation Unity, Instituto Nacional de Enfermedades Respiratorias Ismael Cosío Villegas (INER), Mexico City 14080, Mexico; boq@rocketmail.com; 5Instituto Nacional de Enfermedades Respiratorias Ismael Cosío Villegas (INER), Mexico City 14080, Mexico

**Keywords:** α-smooth muscle actin, collagen type I, E-cadherin, fibroblast, mesenchymal–epithelial transition, microRNA, epithelial-like cell

## Abstract

In passages above ten and growing very actively, we observed that some human lung fibroblasts cultured under standard conditions were transformed into a lineage of epithelial-like cells (ELC). To systematically evaluate the possible mesenchymal–epithelial transition (MET) occurrence, fibroblasts were obtained from normal lungs and also from lungs affected by idiopathic interstitial diseases. When an unusual epithelial-like phenotypic change was observed, cultured cells were characterized by confocal immunofluorescence microscopy, immunoblotting, immunocytochemistry, cytofluorometry, gelatin zymography, RT-qPCR, and hybridization in a whole-transcript human microarray. Additionally, microvesicles fraction (MVs) from ELC and fibroblasts were used to induce MET, while the microRNAs (miRNAs) contained in the MVs were identified. Pattern-gene expression of the original fibroblasts and the derived ELC revealed profound changes, upregulating characteristic epithelial-cell genes and downregulating mesenchymal genes, with a marked increase of E-cadherin, cytokeratin, and ZO-1, and the loss of expression of α-SMA, collagen type I, and Thy-1 cell surface antigen (CD90). Fibroblasts, exposed to culture media or MVs from the ELC, acquired ELC phenotype. The miRNAs in MVs shown six expressed exclusively in fibroblasts, and three only in ELC; moreover, twelve miRNAs were differentially expressed between fibroblasts and ELC, all of them but one was overexpressed in fibroblasts. These findings suggest that the MET-like process can occur in human lung fibroblasts, either from normal or diseased lungs. However, the biological implication is unclear.

## 1. Introduction

Fibroblasts represent a heterogeneous cell population constituted by several subsets, including classic fibroblasts, myofibroblasts, pericytes, and interstitial contractile cells. These subsets participate dynamically in the turnover of the extracellular matrix during physiological and pathological conditions. Fibroblasts derived from the mesenchyme are characterized by a phenotype with elongated fusiform appearance, migratory behavior, and expression of typical mesenchymal markers, including vimentin, N-cadherin, collagen type I, and matrix metalloproteinase-2 (MMP-2) [1]. Fibroblasts play crucial roles in several physiological processes during development and in adult tissue. These cells also participate in pathological conditions, including inflammatory diseases, metastatic cancers, and fibrotic disorders [1,2,3].

During development, fibroblasts change their mesenchymal phenotype to an epithelial phenotype, in a process defined as the mesenchymal–epithelial transition (MET), which is fundamental in global ontogenetic development [4,5]. MET involves a complex functional phenotypic change from a typical mesenchymal nonpolarized cell to a polarized ELC. In this process, fibroblasts lose their spindle-shaped morphology and migratory capacity. The expression of typical markers, such as vimentin, desmin, α-SMA, N-cadherin, collagen type I and III, and Thy-1 cell surface antigen (CD90), is lost as well [4,5,6]. These features are replaced by polarity and the expression of epithelial markers, such as E-cadherin, tight junction protein 1 (TJP1), also known as zonula occludens protein-1 (TJP1/ZO-1), cytokeratins, type IV and VII collagen, and laminin [7].

In adult tissues, the MET program’s abnormal activation has mainly been associated with metastatic processes [6,7,8]. However, this process has not been described in adult human lung fibroblasts [9,10]. Our laboratory observed that some fibroblasts cultured for several passages under standard conditions spontaneously changed their typical mesenchymal phenotype to an ELC lineage, perhaps through a MET-like process. Therefore, the present study aimed to systematically evaluate the potential occurrence of a MET-like process in lung fibroblasts obtained from healthy subjects and patients with different interstitial lung diseases in vitro, and to obtain insight into the mechanisms involved.

## 2. Materials and Methods

### 2.1. Ethics Statement

The present study was approved by the Ethics, Scientific, and Biosecurity Committees at Instituto Nacional de Enfermedades Respiratorias Ismael Cosío Villegas, where this work was performed under the approved protocol number B10-14.

### 2.2. Cell Culture

Six human lung fibroblast cell lines were randomly selected from 12 lines where a MET-like process had been observed. Three of these lines were derived from normal subjects after brain death (two males and one female: 30, 42, and 54 years old, respectively), and the other three lines were derived from biopsies from patients with different interstitial lung diseases (ILD) (three males: 65, 66, and 68 years old, respectively). Fibroblasts were obtained as previously described [2,11] and cultured in 25 cm^2^ tissue culture flasks using Ham’s F-12 nutrient mixture (F-12) supplemented with 10% FBS, 100 U/mL of penicillin, 100 μg/mL of streptomycin, and 2.5 μg/mL of amphotericin B. All cultures were maintained at 37 °C in a humidified atmosphere of 5% CO_2_. The experiments were performed when cells reached 75% confluence or spontaneously exhibited an epithelial-like phenotype at 10 to 100% predominance than the original fibroblasts. The cells were fixed for immunofluorescence, collected for cytofluorometry, or lysed to obtain cellular extracts for Western blotting or RNA for microarray analysis. The culture media were collected and frozen at −70 °C until subsequent use for gelatin zymography or the extraction and purification of MVs. Subsequent microRNA expression analysis was performed.

### 2.3. Immunofluorescence and Image Acquisition

Fibroblasts and ELC (1 × 10^4^ cells/cm^2^) were grown on coverslips, fixed with methanol/acetone at −20 °C (2 min each), and then washed and maintained in PBS at 4 °C. After blocking with a commercial reagent (Universal Blocking Reagent; Biogenex), the cells were incubated overnight with a mouse antihuman α-SMA (Cat. No. A2547-0.5ML Lot# 073M4761; Sigma, St. Louis, MO, USA; 1:100 dilution) and rabbit antihuman E-cadherin (H-108, sc-7870 Lot# K0315; Santa Cruz Biotechnology, Santa Cruz, CA, USA; 1:25 dilution) at 4 °C, and then washed twice with PBS-Tween 0.5% and incubated with secondary antibodies against mouse (DyLight-488) and rabbit (DyLight-549; 1:250 dilution), for one hour at room temperature. The coverslips were rinsed twice with PBS and mounted with medium containing DAPI to stain cell nuclei (Ultra Cruz Mounting Medium; Santa Cruz Biotechnology, Santa Cruz, CA, USA). Images were obtained with a laser-scanning microscope (Olympus, FsV1000) integrated with an inverted microscope IX81 with UPlanFLN 40 × NA 1.3 objective (Olympus) using Fluoview software. Images were processed and merged using Fiji software.

### 2.4. Immunocytochemistry and Image Acquisition

With the aim of characterizing the presence of the typical phenotype of fibroblast and cells that changed to ELC, cells (passage = 4, *n* = 3) were grown on coverslips (1 × 10^4^ cells/cm^2^), fixed with methanol/acetone at −20 °C (2 min each), and then washed and kept in PBS at 4 °C. After blocking with a commercial reagent (Universal Blocking Reagent; Biogenex), the cells were incubated overnight with mouse antihuman antibodies: Thy-1 cell surface antigen (CD90) (Sc-59398, Santa Cruz Biotechnology; 1:200 dilution); rabbit anti-PDGFRβ antibody (ab32570; Abcam: 1:600 dilution); rabbit antivimentin antibody (ab92547; Abcam; 1:4000 dilution); anticollagen-1 (SC-8784; Santa Cruz Biotechnology, Santa Cruz, CA, USA; 1:20 dilution); mouse antipancytokeratin antibody (SC-58826; Santa Cruz Biotechnology, CA, USA; 1:6000 dilution); antimouse IgG antibody PE-conjugated (405307; BioLegend) and antirabbit IgG antibody APC-conjugated (F0111; R&D). Each antibody was incubated at 4 °C overnight. After washing with PBS, the coverslips were incubated with the appropriate biotinylated secondary antibodies (antimouse or antigoat) (Vector Laboratories, Burlingame, CA, USA) for 30 min at room temperature. After washing with PBS, the reaction was amplified and revealed with the Vectastain Elite ABC kit (Vector Laboratories). The coverslips were rinsed twice with PBS and mounted with medium containing (Santa Cruz Mounting Medium, Santa Cruz Biotechnology, Santa Cruz, CA, USA). Secondary anti IgG antimouse alone was used as a negative control. Cell nuclei were counterstained with hematoxylin. Images were obtained with a laser-scanning microscope (Olympus, FsV1000).

### 2.5. Immunoblotting

The expression of some specific mesenchymal and epithelial biomarkers was evaluated in the original fibroblasts and fibroblasts that acquired the epithelial-like phenotype. The cells were lysed with RIPA buffer (Sigma-Aldrich Corporation, St. Louis, MO, USA) at 4 °C, and the protein concentration was determined with Bradford assay reagent (Bio-Rad Laboratories Inc., Hercules, CA). Protein extracts were mixed (*v*/*v*) with Laemmli buffer and fractionated under reducing conditions on 10% SDS-PAGE. The gels were transferred to nitrocellulose membranes (Bio-Rad Laboratories Inc., Hercules, CA, USA) and blocked with fat-free milk (5% in TBS/0.1% tween). The membranes were incubated overnight at 4 °C with the corresponding primary antibody: anti-α-SMA (Sigma, St. Louis, MO, USA; Cat. No. A-2547; Lot# 033M4768; 1:500 dilution), anti-α1 (I) collagen (Cat. No. SC-8783 COL1A1 (L-19); Lot# E0113; Santa Cruz Biotechnology, Santa Cruz, CA, USA; 1:200 dilution), anti-E-cadherin (BioGenex, San Ramon, CA, USA; Cat. No. AM390-5M; Lot# MU3900116; 1:100 dilution), antipancytokeratin clone C-11 cytokeratins 4, 5, 6, 8, 10, 13, and 18 (Sigma, St. Louis, MO, USA; Cat. No. C2931-0.2ML; Lot# 023M4832; 1:200 dilution), antizonula occludens protein-1 (TJP1/ZO-1; Sc-10804; Santa Cruz Biotechnology, Santa Cruz, CA, USA; 1:200 dilution), and anti-β-tubulin (D-10; Cat. No. SC-5274; Lot# E2516; Santa Cruz Biotechnology, Santa Cruz, CA, USA; 1:200 dilution). MVs were characterized with the antihuman CD63 antibody (Cat. SBI #EXOAB-CD63A-1; rabbit antihuman) with goat antirabbit HRP secondary antibody (System Biosciences; Palo Alto, CA, USA, 1:500 dilution). The membranes were incubated 1 h at room temperature with a specific secondary antibody coupled to peroxidase antirabbit, antigoat or antimouse (Cat. No. 31402; Zymed-Thermo Fisher Sci., CA, USA; 1:200 dilution) and subsequently washed four times for 10 min with PBS-T (0.05% Tween). Immunodetection was performed by chemiluminescent reaction (West Pico Chemiluminescent Substrate, Thermo Fisher Sci.). Images were processed using the Molecular Imager ChemiDoc XR. Image Lab v6.0 software was used for densitometric analysis (Bio-Rad Laboratories Inc., Hercules, CA, USA). The results were normalized against β-tubulin.

### 2.6. Flow Cytometry Analysis

Fibroblasts and their corresponding ELC (500,000 cells from each one) were stained with antihuman E-cadherin antibody (Pe-Cy5.5; clone 67A4; Cat. No. 92121; Biolegend, San Diego, CA, USA; 1:200 dilution), and antihuman CD90 antibody (APC; Sc-59398, Santa Cruz Biotechnology; 1:200 dilution). The cells were washed with PBS and fixed with 1% paraformaldehyde. Fixed cells were incubated with the specific antibodies for 1 h at 4 °C. The stained cells were acquired in a FACSAria Flow Cytometer. Dot-plot images were obtained. The obtained data were analyzed with the FACSDiva software version 6.1 (BD Biosciences, San Jose, CA, USA).

Additionally, in order to characterize that the fibroblasts that became ELC were indeed only fibroblasts, similar primary fibroblasts (passage = 4, *n* = 3) to those that became ELC, were cultured under standard conditions to characterize the presence of typical fibroblast biomarkers, and pancytokeratin, which is specific to epithelial cells. Fibroblast suspensions were washed with blocking solution (cell staining buffer, BioLegend Cat. No. 420201) followed by fixation with 4% paraformaldehyde during 10 min at room temperature. Cells were washed and incubated with FACS permeabilizing solution (BD Bioscience, Cat. No. 340973) for 10 min at room temperature in the dark. After washing the cells, the appropriate primary antibodies were added in FACS permeabilizing solution and incubated for 30 min at 4 °C. After washing the cells twice, the appropriate fluorochrome-conjugated secondary antibodies were added and incubated for 30 min at 4 °C. Mouse antipancytokeratin antibody (sc-58826; Santa Cruz); rabbit anti-PDGFRβ antibody (ab32570; Abcam); rabbit antivimentin antibody (ab92547; Abcam); antimouse IgG antibody PE-conjugated (405307; BioLegend,) and antirabbit IgG antibody APC-conjugated (F0111; R&D) were used. Cells were analyzed by flow cytometry (FASCanto II, Becton Dickinson), and data were analyzed by Flow Jo (TriStar).

### 2.7. Gelatin Zymography

Gelatin zymography is a powerful and useful technique for measuring the relative amounts of active and inactive gelatinases (zymogens) in aqueous samples by measuring gelatin hydrolysis. The enzymes MMP-2 (gelatinase A) and MMP-9 (gelatinase B) are fractionated in SDS-PAGE gels, to be activated by denaturation and detected by including gelatin as a substrate in the gel. The gelatin hydrolysis produces two or three white bands, for each zymogen or active form, after staining with Coomassie blue [12].

MMP-2 and MMP-9 gelatinase activities were assayed, as previously described [12]. Briefly, the samples were mixed with an equal volume of 2× sample buffer, resolved under nonreducing conditions on 7.5% SDS-PAGE containing 1 mg/mL of gelatin as a substrate (Sigma, St. Louis, MO, USA; Cat. No. G-8150; Lot. 63H06591). The conditioned media from U2-OS human cells were used as MMP-9 positive control.

### 2.8. Microarray Analysis

RNA was obtained after lysis and purification with Trizol reagent (Invitrogen Life Technologies, Grand Island, NY, USA), according to the manufacturer’s instructions, and hybridized into Whole-Transcript Microarrays (Human Gene 1.0 ST, Affymetrix, Santa Clara, CA, USA) as indicated by the Affymetrix protocol. Microarray data were preprocessed using R and Bioconductor. Raw intensity values were background corrected, log2 transformed, and then RNA normalized, using algorithms coded in the “oligo” package in Bioconductor. To identify differentially expressed genes, we fit a linear model using the Limma package [13]. Finally, lists of differentially expressed genes for each of three comparisons were generated, selecting the statistically significant genes according to two summary statistics: the log fold-change and the B-statistic. The fold-change cutoff was equal to 1, combined with a confidence measure based on the B-statistic greater than zero. 

The data obtained in this publication were deposited in the NCBI Gene Expression Omnibus and are accessible through GEO Series accession number GSE107677. (https://www.ncbi.nlm.nih.gov/geo/query/acc.cgi?acc=GSE107677 (accessed on 31 January 2021)).

### 2.9. Ingenuity Pathway Analysis (IPA)

To identify networks differentially regulated between the two groups, differentially expressed genes (two-fold or higher) were selected from the lists above to draw canonical pathways and related functions with Ingenuity Pathway Analysis Software (http://www.ingenuity.com/ (accessed on 31 January 2021)) (*p*-value < 0.05).

### 2.10. Isolation of Macrovesicles (MVs)

To assess whether MVs obtained from ELC induced the phenotypic change in human normal lung fibroblasts (passage 4), two different fibroblast cultures—one culture from normal human lung and one culture from idiopathic pulmonary fibrosis (IPF; passage 4)—were treated for eight to ten days (~4 media changes) until ELC were observed (passage 4). The technique used in obtaining the MVs was based on a commercial kit (ExoQuick-TC™ Exosome Precipitation Solution; Cat. No. EXOTC50A-1; System Biosciences), following the manufacturer’s recommendations. Secreted MVs were recovered by differential centrifugation from conditioned media obtained from fibroblasts grown 24 h without serum. Briefly, 80 mL of collected media were centrifuged at 1500× *g* for 10 min at 4 °C, followed by centrifugation at 3200× *g* for 10 min at 4 °C min, to remove dead cells and cell debris. The supernatants were recovered and centrifuged again at 20,000× *g* at 4 °C for 2 h to isolate the pellet with MVs [13,14]. The purity of the MVs obtained was confirmed by immunoblotting with CD-63 specific antibodies. Subsequently, the expression of E-cadherin was assessed by Western blotting, and phase-contrast microscopy images were obtained to examine the morphologic change from fibroblast to ELC. 

### 2.11. Quantitative RT-PCR of MicroRNAs from MVs

Purified MVs from two different ELC and two fibroblast cell lines were analyzed for microRNA (miRNA) expression using microRNA Ready-to-Use PCR panels (Exigon Inc., MA, USA) according to the manufacturer’s instructions [15]. Briefly, RNA was isolated from MVs using the SeraMir Exosome RNA Amplification Kit (SBI Inc., Los Angeles, CA, USA). The resulting RNA (40 ng) was reverse-transcribed with the polyadenylation and cDNA synthesis kit (Universal cDNA Synthesis Kit II) and then analyzed in the qPCR panels. The assays were performed in triplicate.

### 2.12. Statistical Analysis

The results are presented as the means ± SD of at least three independent experiments. Statistical analyses were analyzed using Student’s t-test; a *p* < 0.05 value was considered significant.

## 3. Results

### 3.1. Spontaneous Changes in Human Lung Fibroblasts in Culture

While performing conventional human lung fibroblast cell culture under standard conditions, some cells unexpectedly began to change their microscopic appearance. Active cell growth continued until all original fibroblasts were replaced by a cell population showing phenotypes similar to epithelial cells, characterized by cell–cell contacts, focal accumulation, and a cobblestone shape. This change typically occurred at passages ten or greater. This sequential change from fibroblasts to ELC is documented in Figure 1 by phase-contrast microscopy.

### 3.2. ELC Overexpresses Epithelial Biomarkers and Downregulates Mesenchymal Biomarkers

To corroborate whether the morphological–phenotypic change corresponded to ELC, we explored the expression of specific biomarkers of mesenchymal and epithelial cells by immunoblotting. Mostly, we focused on the expression of α-SMA and collagen type I as characteristic markers of mesenchymal cells, and E-cadherin, cytokeratins (pancytokeratins 4, 5, 6, 8, 10, 13, and 18), and TJP1/ZO-1 that featured the epithelial phenotype. ELC showed a strong expression of the epithelial biomarkers and virtually no mesenchymal markers, while the original culture of fibroblasts (passage = 8) from which the ELC were derived showed mesenchymal but not epithelial biomarkers (Figure 2). This finding was corroborated by confocal microscopy, where α-SMA was strongly expressed in fibroblasts with virtually no expression in the ELC.

Conversely, after the phenotypic change, α-SMA was absent whereas E-cadherin was strongly expressed. This finding is illustrated in Figure 3 in a sample containing approximately 20% fibroblasts and 80% ELC. The upregulation of E-cadherin, a major biomarker of epithelial cells, was also confirmed by flow cytometry (Figure 4). Thus, while fibroblasts exhibited negligible E-cadherin expression, their derived ELC showed a high content of this glycoprotein (87.6 ± 10.4%; *p* < 0.01) (Figure 4A).

In contrast, Thy-1 cell surface antigen was downregulated in lung fibroblasts after undergoing MET and exhibited the ECL phenotype (Figure 5A,B). Thus, while fibroblasts showed a high percentage of positive cells (88.7 ± 12.0%), this percentage was significantly decreased in the derived ELC (18.9 ± 15.9%; *p* < 0.01). A similar effect was observed in all four cell lines analyzed. A representative image of the cytometry to CD90 is shown in Figure 5C,D.

### 3.3. Increased MMP-9 and Decreased MMP-2 Activities in ELC

MMP-2 but not MMP-9 is expressed in lung fibroblasts [15]. In contrast, lung epithelial cells primarily express MMP-9 [16]. Both enzymes activities were examined by gelatin zymography. Our results demonstrate that the increase in the number of ELC is paralleled by the increase in MMP-9 and the decrease in MMP-2 as the fibroblast content in the culture decreased (Figure 6).

### 3.4. Microarray Analysis Shows Changes in the Pattern of Gene Expression

Using the Affimetrix Whole-Transcript Microarrays platform, we examined the global gene expression levels in human lung fibroblasts and their derived ELC, as illustrated in supplementary microarray (Microarrays data deposited in NCBI’s Gene Expression Omnibus and are accessible through GEO Series, accession number GSE107677; https://www.ncbi.nlm.nih.gov/geo/query/acc.cgi?acc=GSE107677 (accessed on 31 January 2021)). A profound change in the gene expression profile was observed in the ELC when compared with the fibroblasts. The results from the whole-transcript human microarray analysis show that some of the upregulated genes characteristic of the epithelial phenotype observed in the ELC included cadherin 1 (CDH1), cadherin 17 (CDH17), matrix metalloproteinase-7 (MMP7), keratin 8 (KRT8), claudin 1 (CLDN1), neurotrophic receptor tyrosine kinase 3 (NTRK3), mucin 13, cell surface-associated (MUC13), CEA cell adhesion molecule 6 (CEACAM6), microtubule-associated protein 7 (MAP7), amphiregulin (AREG), neuronal cell adhesion molecule (NCAM), and neuronal cell adhesion molecule (NRCAM) (Table 1).

The genes relevant to MET, including collagen type I alpha 2 chain (COL1A2), collagen type III alpha 1 chain (COL3A1), matrix metallopeptidase 2 (MMP2), matrix metallopeptidase 14 (MMP14), TIMP metallopeptidase inhibitor 3 (TIMP3), Thy-1 cell surface antigen (CD90), fibroblast activation protein alpha (FAP), snail family transcriptional repressor 2 (SNAI2), fibromoulin (FMOD), platelet derived growth factor receptor alpha (PDGFRA), and fibroblast growth factor 7 (FGF7), were downregulated (Table 2).

### 3.5. Canonical Pathways Related to MET Are Downregulated in ELC

Subsequent ingenuity pathway analysis of the differentially expressed genes in the ELC compared to fibroblasts (Figure 7A) revealed that the TGF-β and Wnt signaling pathways were the most affected, and both pathways are critical for the epithelial–mesenchymal transition (EMT) and MET processes. As shown in Figure 7B, several transcription factors, including three important master regulators in the EMT process such as SNAIL, TWIST, and zinc-finger E-box-binding (ZEB), were decreased in these pathways, resulting in the increased expression of E-cadherin in ELC.

### 3.6. Secreted MVs from ELC Induce a MET-Like Process in Human Lung Fibroblast

MVs are heterogeneous, membrane-bound sacs, shed from the surface of several cell types, having a size from 100 to 1000 nm, capable of encapsulating and transferring multiple forms of cargo, including proteins, mRNA, and miRNAs, siRNA, and plasmid DNA. MVs have relevant roles in transforming the extracellular environment, intercellular signaling, facilitating cell invasion through cell-independent matrix proteolysis, and modifying genetic expression and cell physiology in the proximal or distal recipient cells. MVs are found in multiple bodily fluids and tissues and could significantly impact future diagnostic and therapeutic strategies [17].

We next explored whether MVs obtained from the ELC could induce this MET-like process in normal lung fibroblasts on the early passage. The results revealed that MVs from ELC induced the expression of E-cadherin (Figure 8A), and the sequential change of morphology (Figure 8B–H), while MVs obtained from normal or IPF fibroblasts cultures (passage 4) had no effect on fibroblasts (Figure 8B–H).

### 3.7. MVs from ELC Showed miRNAs Specific and Downregulated, But One Overexpressed Compared to Fibroblast

MVs are composed of lipids, proteins, and microRNAs (miRNAs). We examined the miRNA expression in the MVs from fibroblasts and ELC to determine whether some of these molecules are involved in MET. MicroRNA expression was evaluated in the MVs of two cell lines by quantitative RT-PCR. A total of six miRNAs were exclusively expressed in MVs from fibroblasts: hsa-miR-10a-3p, hsa-miR-125b-1-3p, hsa-miR-214-5p, hsa-miR-376a-5p, hsa-miR-380-3p, and hsa-miR-1471. Three miRNAs were exclusively expressed in MVs from ELC: hsa-miRNA-29b, a microRNA implicated in MET (see below), hsa-miR-96-5p, and hsa-miR-935. Similarly, twelve miRNAs were differentially expressed between fibroblasts and ELC; eleven were overexpressed in fibroblasts compared with ELC (Table 3), but one of these, hsa-miR-30b-5p, was overexpressed in ELC versus fibroblasts.

### 3.8. Characterization of Human Lung Fibroblast Phenotype 

Finally, once we showed the differences between the original fibroblasts and the ELC resulting from the spontaneous conversion, we proceeded to verify the phenotype of the fibroblasts that are routinely cultured in our laboratory. For this purpose, we stained the cells with antibodies against proteins expressed, specifically in fibroblasts (Thy-1 cell surface antigen, vimentin, type I collagen, PDGFRβ), and with an antibody against pancytokeratin, a specific protein present in epithelial cells but not in fibroblast. As expected, the fibroblasts are positive for fibroblast-markers and negative for epithelial cells, corroborating that they are phenotypically indeed fibroblasts, as shown in Figure 9.

## 4. Discussion

EMT and the opposite MET are crucial processes in ontogenic development to control cell plasticity, postnatal growth, and tissue homeostasis under physiological and pathological conditions [18,19,20,21]. In the lungs, an EMT-like process has been revealed in IPF, the most common and lethal of the idiopathic interstitial pneumonias, where aberrantly activated epithelial cells acquire a mesenchymal-like phenotype [22,23,24,25]. To our knowledge, a MET-like process has not been reported in human lung fibroblasts in vivo. However, a recent study showed that the transfection of primary lung fibroblasts with the miRNA-let-7 induced a decrease in the expression of several mesenchymal markers and an increase in the expression of epithelial markers, such as tight junction protein-1 and keratin 19 [21]. Similarly, MET was induced in other cell types using a variety of factors, including miRNA-147 [26], miRNA-200a [27], or a mix of reprogramming factors [28]. Significantly, bone morphogenic protein-7 induces MET in adult renal fibroblasts, mimicking this protein’s effect on embryonic metanephric mesenchyme to generate epithelium and facilitate tissue regeneration after injury [29].

Our serendipitous observation showed that some human lung fibroblasts spontaneously changed their fusiform-shaped mesenchymal phenotype to an epithelial-like cobblestone-shaped cell lineage. The phenomenon occurs when the cells are in passage greater than ten and exhibiting a very active growth, whereby we evaluated this process in a systematic protocol. We first demonstrated that during this morphologic change, fibroblasts lose the typical mesenchymal proteins α-SMA and collagen type I, with the concomitant increase of epithelial markers, such as E-cadherin and cytokeratins, thus showing significant features of MET in a 6–10-day term. Additionally, the expression of MMP-2 was strongly reduced, whereas MMP-7 and MMP-9, common epithelial cell enzymes, were overexpressed in these ELC. The expression profile observed in the comparative microarray analysis between fibroblasts and their derived ELC also supported a MET-like genotypic change program. Thus, the observed transition of cell cultures of fibroblasts to ELC was characterized by a strong downregulation of typical mesenchymal markers. Simultaneously, we noticed a progressive increase in epithelial markers involved in epithelial cell-cell adhesion [30], (E-cadherin and claudin 1): MMP-7 (commonly expressed in epithelial lung cells [31]), keratin 8 (an ETS transcription factor expressed by epithelial cells to enhance the expression of MMP-9 [32]), and microtubule-associated protein 7, which is essential for cell polarization and epithelial differentiation [33]. In contrast, several genes representative of the mesenchymal phenotype were downregulated, including collagen type I, MMP2, fibronectin 1, CD90, and fibroblast activation protein alpha (FAP), a type II transmembrane serine protease involved in epithelial–mesenchymal interactions during development and tissue repair [34].

Analysis of microarray and IPA canonical pathways revealed a downregulation of TGF-β and Wnt signaling pathways, which are typically inducers of EMT and fibroblast activation, and a decrease in the expression of several transcription factors implicated in EMT/MET, which are known repressors of the epithelial phenotype and activators of the mesenchymal phenotype. ZEB is a transcription factor whose expression is post-transcriptionally repressed by microRNAs from the miR-200 family, which promote MET via the repression of the expression of ZEB1 and ZEB2 [35]. Interestingly, miR-192 (a miR-215 homolog), which also targets and reduces ZEB1 and ZEB2 expression, was upregulated in the ELC compared to fibroblasts, consistent with ZEB downregulation [36].

Importantly, we were able to mimic the MET-like process using MVs but not exosomes from the ELC cultures. Thus, lung fibroblasts stimulated with ELC-derived MVs displayed a strong upregulation of E-cadherin and acquired the epithelial-like phenotype. MVs are heterogeneous membrane-bound sacs that shed from the surface of many cell types. These vesicles play essential roles in intercellular signaling and the cell–extracellular environment under physiological and disease conditions [17]. Notably, the effect of the MVs is determined by vesicle cargo content, which is in turn dependent upon the cell type from which they are shed. In the present study, we focused on the microRNA content of these MVs, and found that several of these microRNAs were present only in fibroblast populations or in the ELC derived from these fibroblasts. Three miRNAs were specifically found in MVs derived from the ELC. Among these molecules, miRNA-29b is significantly reduced in human prostate cancer tissues, and its reintroduction into metastatic prostate cancer cells induces a change in morphology, reduces migration/invasion, and more importantly, results in the upregulation of E-cadherin with the concomitant decrease of N-cadherin, Twist, and Snail [37]; miRNA-29b also participates in the epigenetic regulation between EMT/MET by hepatocyte nuclear factor 4 alpha (HNF4α), maintaining hepatocyte identity through the regulation of the expression of miRNA-29b (but also of miR-29a) by limiting epigenetic modifications in DNA methyltransferases 3A and 3B levels [38]. Similarly, the transfection of miRNA-29b into an invasive cancer cell line from the submaxillary salivary gland reverses EMT gene expression, increases E-cadherin, and decreases N-cadherin and Vimentin [39]. In addition, as observed in total cells, miRNA-192 was also upregulated in the ELC. 

In MVs purified from ELC, hsa-miR-30b-5p was the only miRNA overexpressed. Recent studies have consistently shown that hsa-miR-30b downregulation suppresses EMT, metastasis and invasiveness in hepatocellular carcinoma, renal carcinoma, and non-small-cell lung cancer [40,41,42,43].

Interestingly, hsa-miR-30b inhibits the TGF-β1-induced EMT in hepatocytes by targeting Snail1, through a similar mechanism as described in the present study, where TGF-β1 signaling was inhibited during the MET in lung fibroblasts [44]. However, this mechanism needs further exploration to determine the specific role of this miRNA in the process of MET.

Finally, once we showed the differences between the original fibroblasts with the ELC resulting from the spontaneous conversion, we proceeded to verify the typical characteristics in fibroblasts. For that, we analyzed by immunocytochemistry and cytofluorometry using specific antibodies for fibroblasts or epithelial cells (antipancytokeratin). Our finding showed that fibroblasts are positive for Thy-1 cell surface antigen, vimentin, type I collagen, PDGFRβ, and negativity to pancytokeratin, which corroborates that they are indeed fibroblasts.

## 5. Conclusions

The data presented here indicate that a MET-like process may spontaneously occur in human lung fibroblasts in vitro when these are in passages higher than ten and exhibit very active growth. This process may be a rare phenomenon occurring in lung fibroblasts, which might also be inherent to these cells since MET was observed, regardless of whether the cells were derived from normal or diseased tissues. The biological significance of this spontaneous phenotypic transition of fibroblasts to ELC is presently unknown; however, the factors identified in the present study set the basis for future research in a field that is still poorly explored. 

## Figures and Tables

**Figure 1 biomolecules-11-00378-f001:**
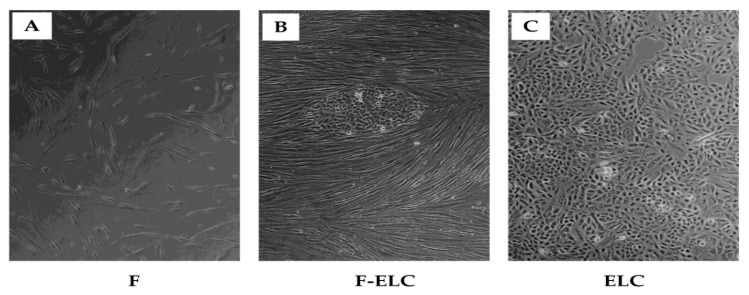
Human lung fibroblasts spontaneously convert into epithelial-like cells (ELC). Serial images of human lung fibroblasts (passage > 10) cultured on coverslips in standard culture technique. Some cells exhibiting an epithelial-like phenotype were observed, and up to 100% of cells showed the epithelial-like morphology could be detected. Cells grown on coverslips were fixed with acetone–methanol (2:1) and observed with phase-contrast microscopy (original magnification ×10). Fibroblasts showed the characteristic phenotype of fibroblasts and/or ELC. Original fibroblasts showing a fusiform and dispersed configuration (**A**). An area with typical fibroblasts and ~20% of ELC exhibiting the polygonal/cobblestone shaped, closely connected, and focally accumulated appearance (**B**). An area covered at 100 % of ELC, where fibroblasts are not apparent (**C**). F: fibroblast; ELC: epithelial-like cell.

**Figure 2 biomolecules-11-00378-f002:**
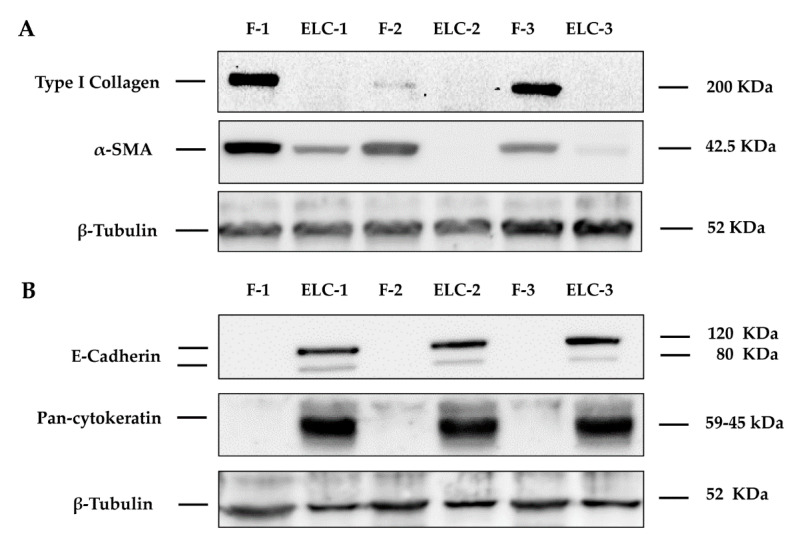
Human lung fibroblast when converted in ELC downregulated α-SMA and collagen type I, and concomitantly upregulated E-cadherin and cytokeratins. The expression of fibroblast and epithelial biomarkers analyzed by Western blotting showed that the change from a spindle to a polygonal/cobblestone shape was consistent with the marked downregulation of α-SMA and collagen type I (**A**), and the overexpression of E-cadherin, pancytokeratin (4, 5, 6, 8, 10, and 13 cytokeratin) (**B**). Three different cell lines were analyzed: two from ILD patients (lines F1/ELC-1; F-2/ELC-2), and two from brain dead patients (F-3/ELC-3; F3/ELC3). β-tubulin was used as a loading control. F: fibroblast; ELC: epithelial-like cell.

**Figure 3 biomolecules-11-00378-f003:**
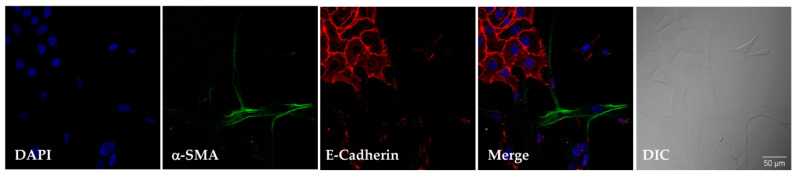
Low α-SMA levels and high E-cadherin levels were detected during the epithelial–mesenchymal transition (EMT)-like process of fibroblasts by confocal immunofluorescence microscopy. Fibroblasts and ELC were grown on coverslips, fixed with paraformaldehyde, and stained with α-SMA and E-cadherin antibodies, followed by incubation with fluorescent dye-tagged secondary antibodies. Images show a sample containing approximately 20% fibroblasts and 80% ELC. DAPI was used to stain nuclei. α-SMA (green), E-cadherin (red), and nuclei (blue). Stained cells were visualized by confocal fluorescence microscopy (original magnification ×40). Differential interference contrast (DIC) is also shown.

**Figure 4 biomolecules-11-00378-f004:**
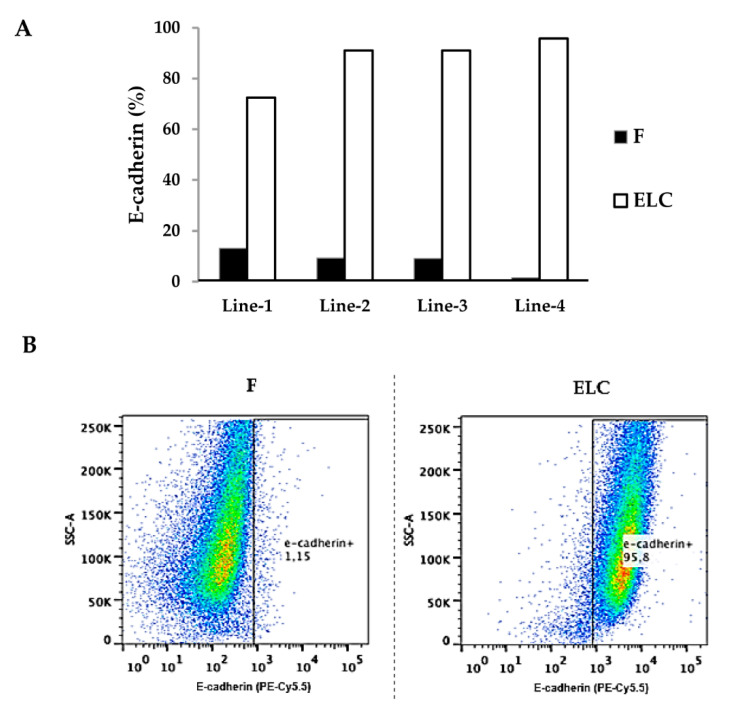
E-cadherin was increased in ELC derived from human lung fibroblasts. The fibroblast-rich culture plates obtained from different passages of the same cell line were compared with the correspondent ELC. The cells were fixed, labeled with an E-cadherin (CD324) antibody (PE-Cy5.5), and analyzed by cytofluorometry. Percentage of cells expressing E-cadherin in four different fibroblast and ELC lines (**A**): two were obtained from ILD patients (Line-1 and Line-2), and two from brain dead patients (Line-3 and Line-4). Representative scatter dot-plot images of flow cytometry for E-cadherin in fibroblasts and ELC (**B**). F: fibroblast; ELC: epithelial-like cell.

**Figure 5 biomolecules-11-00378-f005:**
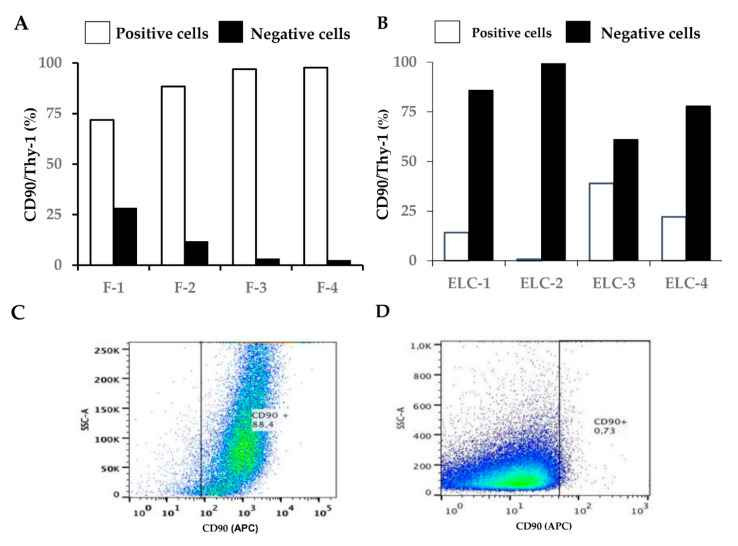
Thy-1 cell surface antigen (CD90) was downregulated in lung fibroblasts after undergoing mesenchymal–epithelial transition (MET) and exhibited the ELC phenotype. Cells were fixed, labeled with a Thy-1 cell surface antigen antibody (APC; Sc-59398, Santa Cruz Biotechnology; 1:200 dilution) and analyzed by cytofluorometry. The percentage of fibroblasts and the derived ELC cells expressing Thy-1 cell surface antigen were examined in four different cell lines (**A**,**B**). Four different cell lines were analyzed: two derived from ILD patients (lines F1/ELC-1; F-2/ELC-2), and two from brain dead patients (F-3/ELC-3; F3/ELC3). F: fibroblast; ELC: epithelial-like cell. C and D: Representative scatter dot-plot images of flow cytometry for CD90 in fibroblasts (**C**) and epithelial-like cell (**D**).

**Figure 6 biomolecules-11-00378-f006:**
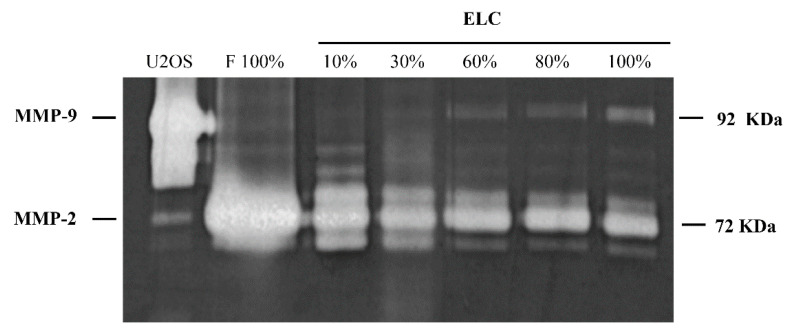
A gradual increase in the MMP-9 activity with a concomitant decrease in the MMP-2 activity is observed as more fibroblasts undergo MET in culture. Basal analysis of gelatinase activities analyzed by zymography in whole culture media from fibroblasts cultures containing different percentages of ELC (10–100%). Culture media from U2OS cells (epithelial cell line) was used as a control. F: fibroblasts; ELC: epithelial-like cells.

**Figure 7 biomolecules-11-00378-f007:**
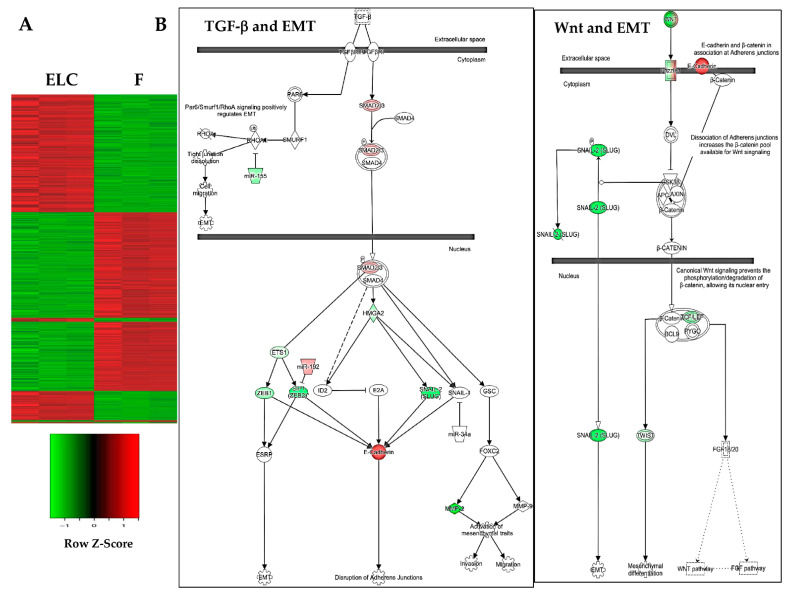
Clustering analysis plot of fibroblasts (F) and derived ELC showing marked changes in pattern-gene expression, with downregulation of canonical TGF-β and Wnt pathways in the ELC derived from fibroblasts. Cluster analysis of fibroblasts and derived ELC. Upregulated and downregulated genes are represented in red and green colors, respectively (**A**). Pathway analysis was performed using ingenuity pathway analysis (IPA) software, as described in the Methods. Transcription factors Snail, Twist, and Zeb were downregulated (green), while E-cadherin was upregulated (**B**).

**Figure 8 biomolecules-11-00378-f008:**
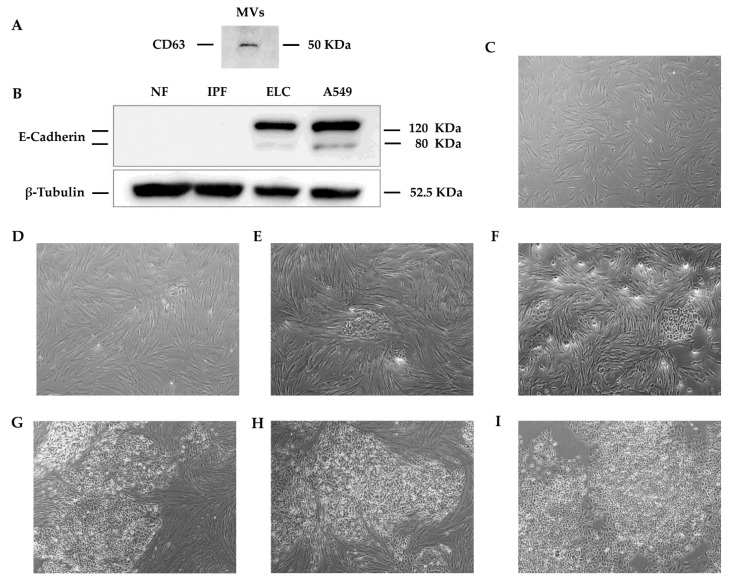
Microvesicles (MVs) obtained from ELC induce the change to an epithelial-like phenotype in human normal lung fibroblasts. Representative immunoblot image to the CD63 protein in MVs purified from ECL from an IPF patient (**A**). MVs effect on E-cadherin expression analyzed by Western blotting: MVs purified from culture media obtained from human normal lung fibroblasts (NF; passage 4), IPF fibroblasts (IPF; passage 4), and ELC (passage 4). Total protein extract from A549 epithelial cell line was used as a positive control (**B**). Serial images of human lung fibroblasts incubated with MVs converted in ELC, observed with phase-contrast microscopy (original magnification ×10): area with 100% of normal lung fibroblasts (**C**). Image captured after 2 days exhibiting ~95% fibroblasts and ~5% ELC displaying a polygonal/cobblestone shaped, closely connected to each other and focally accumulated appearance (**D**). Image obtained after 4 days exhibiting ~90% fibroblasts and ~10% of ELC (**E**). Image acquired after 6 days showing ~65% fibroblasts and ~35% of ELC (**F**). Image taken after 8 days displaying ~40% fibroblasts and ~60% of ELC (**G**). Image obtained after 10 days displaying ~20% fibroblasts and ~80% ELC (**H**). Area covered at ~100% by ELC captured after 12 days of culture, where fibroblasts are not apparent (**I**).

**Figure 9 biomolecules-11-00378-f009:**
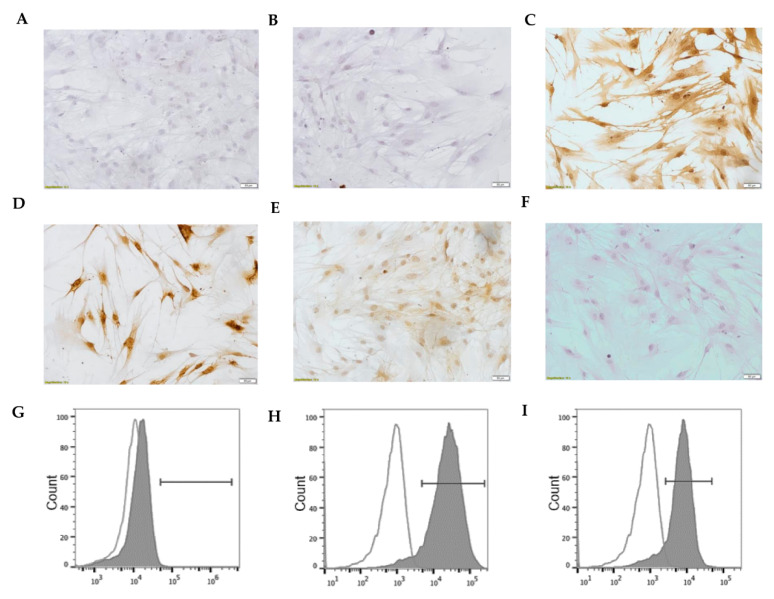
Similar fibroblasts to those converted in ELC showed characteristic biomarkers of fibroblast phenotype but not pancytokeratin. Normal lung fibroblasts (passage = 4; *n* = 3) were grown and seeded on coverslips, fixed, and labeled with antibodies for immunocytochemistry before being analyzed by cytofluorometry. Pancytokeratin (**A**), Thy-1 cell surface antigen (**B**), vimentin (**C**), type I collagen (**D**), PDGFRβ (**E**), secondary anti IgG antimouse alone, used as a negative control (**F**). Cell fluorometric analysis confirmed the absence of pancytokeratin (**G**), and the presence of PDGFRβ (**H**), and vimentin (**I**). Representative flow cytometry histograms showing the expression of pancytokeratin (negative) (**G**), PDGFRβ (positive) (**H**), and vimentin (positive) (**H**). Unshaded histogram corresponds to the staining control with the secondary antibodies alone.

**Table 1 biomolecules-11-00378-t001:** The 100 most upregulated genes expressed by ELC after the MET from human lung fibroblasts are listed. Fold change for each gene is indicated; symbol and gene name are in progressive order of expression.

Fold Change	Symbol	Entrez Gene Name	Fold Change	Symbol	Entrez Gene Name
8.39	AKR1B10	aldo-keto reductase family 1, member B10	4.83	UPK1B	uroplakin 1B
7.74	SLCO1B3	solute carrier organic anion transporter family member 1B3	4.83	ST6GAL2	ST6 beta-galactoside alpha-2,6-sialyltransferase 2
7.72	CYP24A1	cytochrome P450 family 24 subfamily A member 1	4.79	TIMD4	T cell immunoglobulin and mucin domain containing 4
7.5	GPX2	glutathione peroxidase 2	4.77	TM4SF20	transmembrane 4 L six family member 20
7.37	CEACAM6	CEA cell adhesion molecule 6	4.77	CXADR	CXADR Ig-like cell adhesion molecule
7.07	KYNU	kynureninase	4.76	SMOC1	SPARC related modular calcium binding 1
6.97	PON3	paraoxonase 3	4.71	NR5A2	nuclear receptor subfamily 5 group A member 2
6.76	SPP1	secreted phosphoprotein 1	4.67	ANKRD18A	ankyrin repeat domain 18A
6.64	ABCC2	ATP binding cassette subfamily C member 2	4.66	CST1	cystatin SN
6.61	CP	ceruloplasmin	4.62	TBC1D8	TBC1 domain family member 8
6.55	HAVCR1	hepatitis A virus cellular receptor 1	4.59	ACSM3	acyl-CoA synthetase medium chain family member 3
6.44	SLC27A2	solute carrier family 27 member 2	4.56	CLDN1	claudin 1
6.44	TNS4	tensin 4	4.5	ANO5	anoctamin 5
6.42	CNTN1	contactin 1	4.49	PLS1	plastin 1
6.3	HORMAD1	HORMA domain containing 1	4.49	ITGB6	integrin subunit beta 6
6.2	LGSN	lengsin, lens protein with glutamine synthetase domain	4.48	HNF1B	HNF1 homeobox B
6.17	HPGD	15-hydroxyprostaglandin dehydrogenase	4.46	ARHGAP26	Rho GTPase activating protein 26
6.08	CEACAM7	CEA cell adhesion molecule 7	4.46	AKR1C1	aldo-keto reductase family 1 member C1
6.07	KRT81	keratin 81	4.41	ELOVL7	ELOVL fatty acid elongase 7
6.06	HGD	homogentisate 1,2-dioxygenase	4.41	SLPI	secretory leukocyte peptidase inhibitor
6.06	AKR1C3	aldo-keto reductase family 1, member C3	4.41	F5	coagulation factor V
5.87	NTRK3	neurotrophic receptor tyrosine kinase 3	4.37	EHF	ETS homologous factor
5.8	AREG	anphiregulin	4.34	MLPH	melanophilin
5.8	KRTS	keratin 8	4.31	HIST1H2AB	H2A clustered histone 4
5.75	MMP7	matrix metallopeptidase 7	4.31	ELF3	E74 like ETS transcription factor 3
5.7	GCNT3	glucosaminyl (N-acetyl) transferase 3, mucin type	4.3	CEACAM5	CEA cell adhesion molecule 5
5.67	AKR1C2	aldo-keto reductase family 1, member C2	4.25	WDR72	WD repeat domain 72
5.66	TM4SF18	transmembrane 4 L six family member 18	4.23	ITGB4	integrin subunit beta 4
5.66	SLC22A3	solute carrier family 22 member 3	4.23	MAP7	microtubule associated protein 7
5.63	CDH1	cadherin 1	4.22	FGB	fibrinogen beta chain
5.52	CDH17	cadherin 17	4.21	TESC	tescalcin
5.46	PPARGC1A	PPARG coactivator 1 alpha	4.19	NRCAM	neuronal cell adhesion molecule
5.41	HNF1A-AS1	HNF1A antisense RNA 1	4.16	IGFBP1	insulin like growth factor binding protein 1
5.36	GPRIN3	GPRIN family member 3	4.15	OAS1	2’-5’-oligoadenylate synthetase 1
5.31	GABRB3	gamma-aminobutyric acid type A receptor subunit beta3	4.11	CDRT1	CMT1A duplicated region transcript 1
5.31	EREG	epiregulin	4.09	TFAP2A	transcription factor AP-2 alpha
5.31	TSPAN7	tetraspanin 7	4.06	TC2N	tandem C2 domains, nuclear
5.29	ARSE	arylsulfatase E	4.06	POF1B	POF1B actin binding protein
5.2	RAB27B	RAB27B, member RAS oncogene family	4.04	mir622	microRNA 622
5.16	SLC17A3	solute carrier family 17 member 3	4.01	MTUS1	microtubule associated scaffold protein 1
5.13	CYP4F11	cytochrome P450 family 4 subfamily F member 11	3.99	CPS1	carbamoyl-phosphate synthase 1
5.13	SLCO1B7	solute carrier organic anion transporter family member 1B7 (putative)	3.99	FILIP1	filamin A interacting protein 1
5.11	RSPO3	R-spondin 3	3.98	CYP4F3	cytochrome P450 family 4 subfamily F member 3
5.01	FGL1	fibrinogen-like 1	3.98	FZD3	frizzled class receptor 3
5	AREG	amphiregulin	3.98	CHRNA5	cholinergic receptor nicotinic alpha 5 subunit
4.99	ALDH3A1	aldehyde dehydrogenase 3 family member A1	3.97	KRT18	keratin 18
4.98	MUC13	mucin 13, cell surface associated	3.96	KCNJ16	potassium inwardly rectifying channel subfamily J member 16
4.86	CLMN	calmin	3.93	GABRA5	gamma-aminobutyric acid type A receptor subunit alpha5
4.85	CA12	carbonic anhydrase 12	3.93	GRB14	growth factor receptor bound protein 14
4.84	TMEM156	transmembrane protein 156	3.91	INSL4	insulin like 4

**Table 2 biomolecules-11-00378-t002:** The 100 most downregulated genes expressed by ELC after the MET from human lung fibroblasts are listed. Fold change for each gene is indicated; symbol and gene name are in progressive order of expression.

Fold Change	Symbol	Entrez Gene Name	Fold Change	Symbol	Entrez Gene Name
−8.32	SPARC	secreted protein acidic and cysteine rich	−4.93	JAM2	junctional adhesion molecule 2
−7.5	SERPINB2	serpin family B member 2	−4.92	FLNC	filamin C
−7.3	POSTN	periostin	−4.92	COL3A1	collagen type III alpha 1 chain
−7.24	LOX	lysyl oxidase	−4.92	FMOD	fibromodulin
−7.15	DCN	decorin	−4.9	XYLT1	xylosyltransferase 1
−6.95	SULF1	sulfatase 1	−4.88	ENG	endoglin
−6.95	IGFBP5	insulin like growth factor binding protein 5	−4.87	PCDH18	protocadherin 18
−6.88	LUM	lumican	−4.87	MAN1A1	mannosidase alpha class 1A member 1
−6.82	TIMP3	TIMP metallopeptidase inhibitor 3	−4.84	TRPS1	transcriptional repressor GATA binding 1
−6.68	FAP	fibroblast activation protein alpha	−4.83	ZNF737	zinc finger protein 737
−6.51	COL6A3	collagen type VI alpha 3 chain	−4.83	EGR1	early growth response 1
−6.5	LAMA4	laminin subunit alpha 4	−4.81	MIR145	microRNA 145
−6.46	CDH11	cadherin 11	−4.81	ARID5B	AT-rich interaction domain 5B
−6.23	ZNF253	zinc finger protein 253	−4.8	FAM26E	calcium homeostasis modulator family member 5
−6.14	THBS2	thrombospondin 2	−4.77	HAS2	hyaluronan synthase 2
−6.11	BGN	biglycan	−4.76	IRAK3	interleukin 1 receptor associated kinase 3
−6.09	CDH13	cadherin 13	−4.76	LHFPL6	LHFPL tetraspan subfamily member 6
−6.01	INHBA	inhibin subunit beta A	−4.7	PDCD1LG2	programmed cell death 1 ligand 2
−5.96	COL1A2	collagen type I alpha 2 chain	−4.69	TNFRSF11B	TNF receptor superfamily member 11b
−5.94	MMP2	matrix metallopeptidase 2	−4.68	ZNF83	zinc finger protein 83
−5.92	SLIT2	slit guidance ligand 2	−4.67	MMP14	matrix metallopeptidase 14
−5.9	SNAI2	snail family transcriptional repressor 2	−4.67	NDNF	neuron derived neurotrophic factor
−5.87	FGF7	fibroblast growth factor 7	−4.65	TNC	tenascin C
−5.76	LY96	lymphocyte antigen 96	−4.63	ANTXR2	ANTXR cell adhesion molecule 2
−5.76	DPP4	dipeptidyl peptidase 4	−4.6	FBN1	fibrillin 1
−5.64	DDR2	discoidin domain receptor tyrosine kinase 2	−4.57	PTGS1	prostaglandin-endoperoxide synthase 1
−5.61	SERPINF1	serpin family F member 1	−4.55	CRYAB	crystallin alpha B
−5.54	EDIL3	EGF like repeats and discoidin domains 3	−4.53	ITGB3	integrin subunit beta 3
−5.46	GREM1	gremlin 1, DAN family BMP antagonist	−4.51	DSEL	dermatan sulfate epimerase like
−5.45	ELTD1	adhesion G protein-coupled receptor L4	−4.5	PDGFRB	platelet derived growth factor receptor beta
−5.43	SERPING1	serpin family G member 1	−4.5	EFEMP2	EGF containing fibulin extracellular matrix protein 2
−5.4	DDR2	discoidin domain receptor tyrosine kinase 2	−4.50	MMP3	matrix metallopeptidase 3
−5.4	SLFN11	schlafen family member 11	−4.47	KGFLP1	fibroblast growth factor 7 pseudogene 6
−5.38	IFITM3	interferon induced transmembrane protein 3	−4.46	LOC100288114	Uncharacterized
−5.36	DOK5	docking protein 5	−4.46	ANGPTL2	angiopoietin like 2
−5.34	CDK14	cyclin dependent kinase 14	−4.45	IFITM1	interferon induced transmembrane protein 1
−5.31	IL6	interleukin 6	−4.45	ZFPM2	zinc finger protein, FOG family member 2
−5.29	PDGFRA	platelet derived growth factor receptor alpha	−4.44	ENPP2	ectonucleotide pyrophosphatase/phosphodiesterase 2
−5.24	GLT8D2	glycosyltransferase 8 domain containing 2	−4.43	EDNRA	endothelin receptor type A
−5.23	PTGIS	prostaglandin I2 synthase	−4.41	OLFML3	olfactomedin like 3
−5.21	KIAA1199	cell migration inducing hyaluronidase 1	−4.37	ADAMTS6	ADAM metallopeptidase with thrombospondin type 1 motif 6
−5.18	PRRX1	paired related homeobox 1	−4.36	EPB41L3	erythrocyte membrane protein band 4.1 like 3
−5.14	WNT5A	Wnt family member 5A	−4.34	PDE5A	phosphodiesterase 5A
−5.1	PXDN	peroxidasin	−4.31	MFAP4	microfibril associated protein 4
−5.06	FLRT2	fibronectin leucine rich transmembrane protein 2	−4.31	CRISPLD2	cysteine rich secretory protein LCCL domain containing 2
−5.06	CDH6	cadherin 6	−4.3	NEXN	nexilin F-actin binding protein
−5.01	NID1	nidogen 1	−4.29	SNORD113-4	small nucleolar RNA, C/D box 113-4
−4.97	PAMR1	peptidase domain containing associated with muscle regeneration 1	−4.28	DNAJC15	DnaJ heat shock protein family (Hsp40) member C15
−4.97	MME	membrane metalloendopeptidase	−4.24	ZEB2	ZEB2 antisense RNA 1
−4.94	COL1A1	collagen type I alpha 1 chain	−4.23	FN1	fibronectin 1

**Table 3 biomolecules-11-00378-t003:** The miRNAs overexpressed in MVs derived from fibroblasts compared with those from ELC. Fold changes are indicated (FC). F: fibroblast; ELC: epithelial-like cell.

F vs. ELC	FC
hsa-let-7a-5p	3.1
hsa-let-7b-5p	66.5
hsa-let-7c	4.4
hsa-let-7d-5p	19.7
hsa-let-7e-5p	5.2
hsa-miR-21-3p	8.1
hsa-miR-23a-3p	3.0
hsa-miR-23b-3p	4.9
hsa-miR-24-2-5p	2.2
hsa-miR-26a-5p	49.4
hsa-miR-27a-3p	16.8
**ELC vs. F**	
hsa-miR-30b-5p	2.4

## Data Availability

All data supporting reported results in this study can be found in archives of corresponding author.

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
