# Peer review of "Mesenchymal–Epithelial Transition in Fibroblasts of Human Normal Lungs and Interstitial Lung Diseases"

_biomolecules, 2021, doi:10.3390/biom11030378_

Round 1
Reviewer 1 Report
Becerril et al. observe that human lung fibroblasts can acquire an epithelial-like phenotype after more than ten passages in culture, and they adopt different techniques to prove the occurrence of Mesenchymal-to-Epithelial Transition (MET). They show an upregulation of epithelial markers (E-cadherin and pan-cytokeratin) and downregulation of mesenchymal markers (aSMA -THY) in epithelial-like cells (ELC) compared to fibroblasts, by Western blot, ICC, and flow cytometry. By zymography, they measure the activities of the gelatinases MMP-9 (higher in epithelial cells) and MMP2 (higher in endothelial cells). They show changes in gene expression by microarray profiling. Finally, they observe that microvesicles (MV) obtained from the conditioned media of ELC cultures were sufficient to induce epithelial conversion of fibroblasts and they analyze the differential expression of microRNAs in MV derived from fibroblasts versus those derived from ELC. In summary, different evidence supports the acquisition of an epithelial-like phenotype, though the frequency and the physiological significance of this phenomenon are not clear.
Main comments:
- In the methods, the authors state that “Six human lung fibroblast cell lines were randomly selected from 12 lines where an MET-like process had been observed”. To have a better idea of the frequency, please clarify how many of the examined cell lines never show a similar MET process.
- 3 of the cell line are obtained from brain dead patients, 3 from IPF patients. 3 cell lines are shown in figure 2, 4 lines in figures 4-5. Please specify what is the difference and the source of these cell lines (brain dead or IPF patients).
- In figure 2 please reduce the contrast/brightness adjustment of the western blot images. Some of the bands in the aSMA plots appear to be white (see ELC samples and extra higher band)
-Make sure the scale bar is correct in figure 3
- If I understand correctly, in most of the experiments, the authors compare ELC and Fibroblasts from the same cell lines (fig 1-7), cell lines 1-3, or 1-4. If so, did they selectively sort ELC and fibroblasts from the same culture dish? How? Were these cell lines selected because of a particularly high frequency of ELC? Based on figure 4, the frequency of fibroblasts in lines 1-4 is very low (1-15%?), but the FACS plots in 4B show a similar density. Would it be possible to know the number of fibroblasts and ELC FACS-analyzed in figure 4B and figure 5? The FACS plot in 4B that is supposed to be for Fibroblasts is labeled Comp-PE-Cy5-A, is that the compensation control rather than the actual sample?
- Table 3, the upregulated miRNAs were supposed to be in italic bold, but the two lists (F, ELC) are identical.
Minor comments:
- Follow the HGNC guidelines for human gene and protein names across the text.
- I highlighted in yellow some typos and passages that are not very clear.

Author Response
RESPONSE TO REVIEWER 1 REQUESTS:
Journal Title: Biomolecules
Article Title: ´´Mesenchymal–Epithelial Transition in Fibroblasts of Human Normal Lungs and Interstitial Lung Diseases ´´.
Thanks a lot to editors and reviewers for commenting on our manuscript.
Following the reviewer’s suggestions, we made several changes to the manuscript, which were written in red.
Below is our point-by-point response to the reviewer's comments and the actions taken to amend the manuscript, indicating the page where the change appears, as well as the new references as they appear in the new manuscript. We hope that the revised manuscript is acceptable to be published in the prominent journal Biomolecules.
Reviewer’s comments:
Comments to the Author
Reviewer 1
Open Review
(x) I would not like to sign my review report
( ) I would like to sign my review report
English language and style
( ) Extensive editing of English language and style required
(x) Moderate English changes required
( ) English language and style are fine/minor spell check required
( ) I don't feel qualified to judge about the English language and style
Yes Can be improved Must be improved Not applicable
Does the introduction provide sufficient background and include all relevant references?
(x) ( ) ( ) ( )
Is the research design appropriate?
( ) (x) ( ) ( )
Are the methods adequately described?
( ) (x) ( ) ( )
Are the results clearly presented?
( ) (x) ( ) ( )
Are the conclusions supported by the results?
( ) (x) ( ) ( )
Comments and Suggestions for Authors
Becerril et al. observe that human lung fibroblasts can acquire an epithelial-like phenotype after more than ten passages in culture, and they adopt different techniques to prove the occurrence of Mesenchymal-to-Epithelial Transition (MET). They show an upregulation of epithelial markers (E-cadherin and pan-cytokeratin) and downregulation of mesenchymal markers (SMA-THY) in epithelial-like cells (ELC) compared to fibroblasts, by Western blot, ICC, and flow cytometry. By zymography, they measure the activities of the gelatinases MMP-9 (higher in epithelial cells) and MMP2 (higher in endothelial cells). They show changes in gene expression by microarray profiling. Finally, they observe that microvesicles (MV) obtained from the conditioned media of ELC cultures were sufficient to induce epithelial conversion of fibroblasts and they analyze the differential expression of microRNAs in MV derived from fibroblasts versus those derived from ELC. In summary, different evidence supports the acquisition of an epithelial-like phenotype, though the frequency and the physiological significance of this phenomenon are not clear.
Submission Date
31 January 2021
Date of this review
07 Feb 2021 05:00:56
Main comments:
- In the methods, the authors state that "Six human lung fibroblast cell lines were randomly selected from 12 lines where a MET-like process had been observed". To have a better idea of the frequency, please clarify how many of the examined cell lines never show a similar MET process.
RESPONSE: Thanks for your observation. This is a good point, the ELC-like phenomenon was observed in each of the 12 cell strains, specifically when these cells were in passage> 10 and exhibited a very active growth, actually the phenomenon occurred similarly in all of them.
- 3 of the cell line are obtained from brain dead patients, 3 from IPF patients. 3 cell lines are shown in figure 2, 4 lines in figures 4-5. Please specify what is the difference and the source of these cell lines (brain dead or IPF patients).
RESPONSE: This is a good point, now is write in the legend of figures 3, 4, 5, specifically what was the origin of the fibroblast cell lines used in the analysis.
Legends to Figures 2, 4, and 5 were rewritten.
Figure 2. Human lung fibroblast when converted in ELC downregulated α-SMA and collagen type I, and concomitantly were upregulate E-cadherin and cytokeratins. The expression of fibroblast and epithelial biomarkers analyzed by western blotting showed that the change from a spindle to a polygonal/cobblestone shape was consistent with the marked downregulation of α-SMA and collagen type I (A), and the overexpression of E-cadherin, pan-cytokeratin (4, 5, 6, 8, 10 and 13 cytokeratin) (B). Three different cell lines were analyzed: two from ILD patients (lines F1/ELC-1; F-2/ELC-2), and two from brain dead patients (F-3/ELC-3; F3/ELC3). β-tubulin was used as a loading control. F: fibroblast; ELC: epithelial-like cell.
Figure 4. E-cadherin was increased in ELC derived from human lung fibroblasts. The cells were fixed, labeled with an E-cadherin (CD324) antibody (PE-Cy5.5), and analyzed by cytofluorometry. Percentage of cells expressing E-cadherin in four different fibroblast and ELC lines (A): two were obtained from ILD patients (Line-1 and Line-2), and two from brain dead patients (Line-3 and Line-4). Representative scatter dot-plot images of flow cytometry for E-cadherin in fibroblasts and ELC (B). F: fibroblast; ELC: epithelial-like cell.
Figure 5. Thy-1 cell surface antigen (CD90) was downregulated in lung fibroblasts after undergoing MET and exhibited the ELC phenotype. Cells were fixed, labeled with a Thy-1 cell surface antigen antibody (APC; Sc-59398, Santa Cruz Biotechnology; 1:200 dilution) and analyzed by cytofluorometry. A and B: The percentage of fibroblasts and the derived ELC cells expressing Thy-1 cell surface antigen, was examined in four different cell lines. Four different cell lines were analyzed: two derived from ILD patients (lines F1/ELC-1; F-2/ELC-2), and two from brain dead patients (F-3/ELC-3; F3/ELC3). F: fibroblast; ELC: epithelial-like cell. C and D: Representative scatter dot-plot images of flow cytometry for CD90 in fibroblasts (C) and epithelial-like cell (D).
- In figure 2 please reduce the contrast/brightness adjustment of the western blot images. Some of the bands in the aSMA plots appear to be white (see ELC samples and extra higher band)
RESPONSE: Thanks for the observation. It is true, the image of a-SMA is overexposed, by this fact it was replaced for other similar images, from other experiments performed in the same cell, but with a lower exposure range, which represents the same result. We hope this Figure 2 would be adequate.
-Make sure the scale bar is correct in figure 3
RESPONSE: Thanks for the observation, the bar size was revised in the confocal microscope and the correct size is 50 micrometers, which is now corrected in Figure 3.
- If I understand correctly, in most of the experiments, the authors compare ELC and Fibroblasts from the same cell lines (fig 1-7), cell lines 1-3, or 1-4. If so, did they selectively sort ELC and fibroblasts from the same culture dish? How? Were these cell lines selected because of a particularly high frequency of ELC? Based on figure 4, the frequency of fibroblasts in lines 1-4 is very low (1-15%?), but the FACS plots in 4B show a similar density. Would it be possible to know the number of fibroblasts and ELC FACS-analyzed in figure 4B and figure 5? The FACS plot in 4B that is supposed to be for Fibroblasts is labeled Comp-PE-Cy5-A, is that the compensation control rather than the actual sample?
RESPONSE: thanks for these observations; Figure 4B shows a low frequency of cells expressing E-cadherin in the culture where only fibroblast phenotype is present (left panel). Conversely, the plot that shows a high cadherin level is where ELC is the predominant population (right panel).
We did not perform sorting, the cells were distinguished based on their expression level of E-cadherin, which is specific for epithelial cells but not for fibroblast.
Concerning the origin of the cell in the analysis, these were actually from a different culture dish. That is, when we noticed that a fibroblast culture began to change its phenotype to the ELC, we waited for it to conflux the cells were collected for analysis. Fibroblasts were obtained from a similar vial to the same cell line that was transformed into ELC, which was cultured and harvested for comparative analysis.
500,000 were used for flow cytometry of each fibroblast or ELC, so the dot plot image is derived from this number of cells.
The selection of the ELC to study was an eventuality when the MET-like phenomenon was observed.
About Comp-PE-Cy5-A, the legend is wrong, the correct one is E-cadherin (PE-
Cy5.5), which is now written in Figure 4.
- Table 3, the upregulated miRNAs were supposed to be in italic bold, but the two lists (F, ELC) are identical.
RESPONSE: Thanks for the observation, effectively is wrong the label of miRNAs. However, we prefer to modify Table 3 including the fold change for the eleven miRNA overexpressed in MVs derived from fibroblasts compared with ELC. Indicating in the text the only miRNA overexpressed in ELC vs. fibroblasts.
In Section 3.7. were changed the title and the text, including the Fold Changes to the miRNA overexpression in fibroblasts and ELC. Table 3 also was modified.
3.7. MVs from ELC Showed miRNAs Specific and Downregulated, but one Overexpressed Compared to Fibroblast
MVs are composed of lipids, proteins, and microRNAs (miRNAs). We examined the miRNA expression in the MVs from fibroblasts and ELC to determine whether some of these molecules are involved in MET. MicroRNA expression was evaluated in the MVs of two cell lines by quantitative RT-PCR. A total of 6 miRNAs were exclusively expressed in MVs from fibroblasts: hsa-miR-10a-3p, hsa-miR-125b-1-3p, hsa-miR-214-5p, hsa-miR-376a-5p, hsa-miR-380-3p, and hsa-miR-1471. Three miRNAs were exclusively expressed in MVs from ELC: hsa-miRNA-29b, a microRNA implicated in MET (see below), hsa-miR-96-5p and hsa-miR-935. Similarly, twelve miRNAs were differentially expressed between fibroblasts and ELC; eleven were overexpressed in Fibroblasts compared with ELC (Table 3), but one of these, hsa-miR-30b-5p, was overexpressed in ELC versus fibroblasts.
Table 3. miRNAs overexpressed in MVs derived from fibroblasts compared with those from ELC. Fold changes are indicated (FC). F: fibroblast; ELC: epithelial-like cell.
|
F vs ELC |
FC |
|
hsa-let-7a-5p |
3.1 |
|
hsa-let-7b-5p |
66.5 |
|
hsa-let-7c |
4.4 |
|
hsa-let-7d-5p |
19.7 |
|
hsa-let-7e-5p |
5.2 |
|
hsa-miR-21-3p |
8.1 |
|
hsa-miR-23a-3p |
3.0 |
|
hsa-miR-23b-3p |
4.9 |
|
hsa-miR-24-2-5p |
2.2 |
|
hsa-miR-26a-5p |
49.4 |
|
hsa-miR-27a-3p |
16.8 |
|
ELC vs F |
|
|
hsa-miR-30b-5p |
2.4 |
Minor comments:
- Follow the HGNC guidelines for human gene and protein names across the text.
RESPONSE: All names were revised and corrected in the text according to HGNC guidelines, also in whole Tables 1 and 2 as in the whole text. 3.4. Microarray Gene Shows a Deep Change in Pattern-gene Expression.
3.4. Section. Title was changed and the section was rewritten according reviewers suggestions.
3.4. Microarray Analysis Shows Changes in the Pattern of Gene Expression
Using the Affimetrix Whole-Transcript Microarrays platform, we examined the global gene expression levels in human lung fibroblasts and their derived ELC. As illustrated in supplementary microarray [Microarrays data deposited in NCBI's Gene Expression Omnibus and are accessible through GEO Series, accession number GSE107677; https://www.ncbi.nlm.nih.gov/geo/query/acc.cgi?acc=GSE107677]. A profound change in the gene expression profile was observed in the ELC when compared with the fibroblasts. The results from the whole-transcript human microarray analysis show that some of the upregulated genes characteristic of the epithelial phenotype observed in the ELC included cadherin 1 (CDH1), cadherin 17 (CDH17), matrix metalloproteinase-7 (MMP7), keratin 8 (KRT8), claudin 1 (CLDN1), neurotrophic receptor tyrosine kinase 3 (NTRK3), mucin 13, cell surface-associated (MUC13), CEA cell adhesion molecule 6 (CEACAM6), microtubule-associated protein 7 (MAP7), amphiregulin (AREG), neuronal cell adhesion molecule (NCAM), and neuronal cell adhesion molecule (NRCAM) (Table 1). The genes relevant to MET, including collagen type I alpha 2 chain (COL1A2), collagen type III alpha 1 chain (COL3A1), matrix metallopeptidase 2 (MMP2), matrix metallopeptidase 14 (MMP14), TIMP metallopeptidase inhibitor 3 (TIMP3), Thy-1 cell surface antigen (CD90), fibroblast activation protein alpha (FAP), snail family transcriptional repressor 2 (SNAI2), fibromoulin (FMOD), platelet derived growth factor receptor alpha (PDGFRA), and fibroblast growth factor 7 (FGF7), were downregulated (Table 2).
- I highlighted in yellow some typos and passages that are not very clear.
RESPONSE: Thanks for the observation.
Spelling, grammar, and punctuation were revised across the entire manuscript; the changes are indicated in red script.
All these typos and passages are now revised and corrected in the new version of the manuscript. Changes now contained in the main document are next:
The abstract was rewritten according reviewers suggestions. For major clarity to readers.
Abstract. In passages above ten and growing very actively, we observed that some human lung fibroblasts cultured under standard conditions were transformed into a lineage of epithelial-like cells (ELC). To systematically evaluate the possible mesenchymal-epithelial transition (MET) occurrence, fibroblasts were obtained from normal lungs and also from lungs affected by idiopathic interstitial diseases. When it was observed an unusual epithelial-like phenotypic change, cultured cells were characterized by confocal immunofluorescence microscopy, immunoblotting, immunocytochemistry, cytofluorometry, gelatin zymography, RT-qPCR, and hybridization in a whole-transcript human microarray. Also, microvesicles fraction (MVs) from ELC and fibroblasts were used to induce MET, while the microRNAs (miRNAs) contained in the MVs were identified. Pattern-gene expression of the original fibroblasts and the derived ELC, revealed profound changes, upregulating characteristic epithelial-cell genes and downregulating mesenchymal genes, with a marked increase of E-cadherin, cytokeratin, and ZO-1, and the loss of expression of a-SMA, collagen type I, and Thy-1 cell surface antigen (CD90). Fibroblasts exposed to culture media or MVs from the ELC, acquired ELC phenotype. The miRNAs in MVs shown six expressed exclusively in fibroblasts, and three only in ELC; moreover, twelve miRNAs were differentially expressed between fibroblasts and ELC, all of them but one was overexpressed in fibroblasts. These findings suggest that the MET-like process can occur in human lung fibroblasts, either from normal or diseased lungs. However, the biological implication is unclear.
Line 121.
2.4. Immunocytochemistry and Image Acquisition
With the aim of characterizing the presence of the typical phenotype of fibroblast and cells that changed to ELC, cells (passage = 4, n = 3) were grown on coverslips (1×104 cells/cm2), fixed with methanol/acetone at −20°C (2 min each), and then washed and kept
Line 266:
The word seed was changed by seeded… in the legend to figure 1.
Figure 1. Human lung fibroblasts spontaneously convert in ELC. Serial images of human lung fibroblasts (passage >10) cultured on coverslips in standard culture technique. This could be detected some cells exhibiting an epithelial-like phenotype were observed, and up to 100% of cells showed the epithelial-like morphology. Cells grown on coverslips were fixed with acetone-methanol (2:1), and observed with phase-contrast microscopy (original magnification x10). Fibroblasts showed the characteristic phenotype of fibroblasts and/or ELC. Original fibroblasts showing a fusiform, and dispersed configuration (A). An area with typical fibroblasts and ~20 % of ELC exhibiting the polygonal/cobblestone shaped, closely connected, and focally accumulated (B). An area covered at 100 % of ELC, where fibroblasts are not apparent (C). F: fibroblast; ELC: epithelial-like cell.
In Figure 2:
a-SMA image was replaced for other similar obtained from the same experiment in the same cell but with lower exposure.
Legend to Figure 4 was rewritten:
Figure 4. E-cadherin was increased in ELC derived from human lung fibroblasts. The cells were fixed, labeled with an E-cadherin (CD324) antibody (PE-Cy5.5), and analyzed by cytofluorometry. Percentage of cells expressing E-cadherin in four different fibroblast and ELC lines (A): two were obtained from ILD patients (Line-1 and Line-2), and two from brain dead patients (Line-3 and Line-4). Representative scatter dot-plot images of flow cytometry for E-cadherin in fibroblasts and ELC (B). F: fibroblast; ELC: epithelial-like cell.
Line 309: fibroblasts after undergoing… Cambia a. fibroblasts after convert in ELC (Figure 6)
Legend to Figure 6 was rewritten:
Figure 6. A gradual increase in the MMP-9 activity with a concomitant decrease in the MMP-2 activity is observed as more fibroblasts undergoing MET in culture. Basal analysis of gelatinase activities analyzed by zymography in whole culture media from fibroblasts cultures containing different percentages of ELC (10-100%). Culture media from U2OS cells (epithelial cell line) was used as a control. F: fibroblasts; ELC: epithelial-like cells.
3.4. Microarray Gene Shows a Deep Change in Pattern-gene Expression
Line 348: Whole-transcript human microarray analysis shows that some
Line 383 Transcription factors snail, twist, and Zeb
Figure 7. Clustering analysis plot of fibroblasts (F) and derived ELC showing marked changes in pattern-gene expression, with downregulation of canonical TGF-b and Wnt pathways in the ELC derived from fibroblasts. Cluster analysis of fibroblasts and derived ELC. Upregulated and downregulated genes are represented in red and green colors, respectively (A). Pathway analysis was performed using ingenuity pathway analysis (IPA) software, as described in the Methods. Transcription factors Snail, Twist and Zeb were downregulated (green), while E-cadherin was upregulated (B).
Lines 422-424:
Section 3.8. was rephrased
3.8. Characterization of Human Lung Fibroblast phenotype
Finally, once we showed the differences between the original fibroblasts and the ELC resulting from the spontaneous conversion, we proceeded to verify the phenotype of the fibroblasts that are routinely cultured in our laboratory. For this purpose, we stained the cells with antibodies against proteins expressed, specifically in fibroblasts (Thy-1 cell surface antigen, vimentin, type I collagen, PDGFRb), and with an antibody against pan-cytokeratin, a specific protein present in epithelial cells but not in fibroblast. As expected, the fibroblasts are positive for fibroblast-markers and negative for epithelial cells, corroborating that they are phenotypically indeed fibroblasts, as shown in Figure 9.
Cytofluorometry Figure 9.
Legend to figure 9 was rewritten
Figure 9. Similar fibroblasts to those converted in ELC showed characteristic biomarkers of fibroblast phenotype and but not pan-cytokeratin. Normal lung fibroblasts (passage = 4; n = 3) were grown and seeded on coverslips, fixed, and labeled with antibodies for immunocytochemistry before being analyzed by cytofluorometry. Pan-cytokeratin (A), Thy-1 cell surface antigen (B), vimentin (C), type I collagen (D), PDGFRb (E), secondary anti IgG anti-mouse alone, used as a negative control (F). Cell fluorometric analysis confirmed the absence of pan-cytokeratin (G), and the presence of PDGFRb (H), and vimentin (I). Representative flow cytometry histograms showing the expression of pan-cytokeratin (negative) (G), PDGFRb (positive) (H), and vimentin (positive) (H). Unshaded histogram corresponds to the staining control with the secondary antibodies alone.
Line 457:
Our serendipitous observation showed that some human lung fibroblasts spontaneously changed their fusiform-shaped mesenchymal phenotype to an epithelial-like cobblestone-shaped cell lineage. The phenomenon occurs when the cells are in passage greater than ten and exhibiting a very active growth, whereby we evaluated this process in a systematic protocol.
Additionally.
Spelling, grammar, and punctuation were revised across the entire manuscript; the changes are indicated in red script.

Reviewer 2 Report
Becerril et al, describe an interesting phenomenon observed in cultured lung fibroblasts that will be of interest to readers working with lung fibroblasts as well as authors working on MET transition. In the most part, the results are presented well and the conclusions drawn are backed up by the experimental data. I have a few issues that need to be addressed
Figure 6: Could the authors provide more details about the gelatine zymography assay and explain the results in more detail for readers unfamiliar with the assay.
In addition the authors state MMP2 and 9 levels are changed in lung fibroblasts and ELC cells. It would be nice if the authors probe the lysates from figure 1 for MMP2 and 9 to show this phenomenon in these cells.
Figure 8: Could the authors provide western blot or electron microscope data shown efficient isolation of microvesicles. If the authors probe for exosome markers such as TSG101 or CD63 or microvesicle markers such as CD40 then this would be sufficient.
Table 3: The authors state miRNAs that are increased are shown in bold italics but this is not the case. It would be more informative for the authors to shown changes in the miRNA as fold differences. In addition the authors describe miRNA in results section 3.7 that are not present in the data presented in table 3.
Results section 3.8 is difficult to understand and therefore it is difficult to review this section of the manuscript. Could the authors describe this data more clearly.
Author Response
RESPONSE TO REVIEWER 2 REQUESTS:
Journal Title: Biomolecules
Article Title: ´´Mesenchymal–Epithelial Transition in Fibroblasts of Human Normal Lungs and Interstitial Lung Diseases ´´.
Thanks a lot to the editor and reviewer for commenting on our manuscript.
Following the reviewer’s suggestions, we made several changes to the manuscript, which were written in red.
Below is our point-by-point response to the reviewer's comments and the actions taken to amend the manuscript, indicating the page where the change appears, as well as the new references as they appear in the new manuscript. We hope that the revised manuscript is acceptable to be published in the prominent journal Biomolecules.
Reviewer’s comments:
Comments to the Author
Reviewer 2
Open Review
(x) I would not like to sign my review report
( ) I would like to sign my review report
English language and style
( ) Extensive editing of English language and style required
( ) Moderate English changes required
(x) English language and style are fine/minor spell check required
( ) I don't feel qualified to judge about the English language and style
Yes Can be improved Must be improved Not applicable
Does the introduction provide sufficient background and include all relevant references?
(x) ( ) ( ) ( )
Is the research design appropriate?
(x) ( ) ( ) ( )
Are the methods adequately described?
(x) ( ) ( ) ( )
Are the results clearly presented?
(x) ( ) ( ) ( )
Are the conclusions supported by the results?
(x) ( ) ( ) ( )
Comments and Suggestions for Authors
Becerril et al, describe an interesting phenomenon observed in cultured lung fibroblasts that will be of interest to readers working with lung fibroblasts as well as authors working on MET transition. In the most part, the results are presented well and the conclusions drawn are backed up by the experimental data. I have a few issues that need to be addressed
Submission Date
31 January 2021
Date of this review
03 Feb 2021 14:52:01
Figure 6: Could the authors provide more details about the gelatin zymography assay and explain the results in more detail for readers unfamiliar with the assay.
RESPONSE: It is a good point to readers; a new paragraph was annexed in section 2.7. Gelatin Zymography.
Gelatin zymography is an extremely sensitive and useful technique for measuring the relative amounts of active and inactive gelatinases (zymogen) into soluble samples by measuring gelatin hydrolysis. The enzymes MMP-2 (gelatinase A) and MMP-9 (gelatinase B) are fractionated on SDS-PAGE to be activated by denaturation and detected by including gelatin as a substrate in the gel, which is hydrolyzed producing two or three white bands, for each one, zymogen or active forms, after Coomassie blue staining [12].
In addition, the authors state MMP2 and 9 levels are changed in lung fibroblasts and ELC cells. It would be nice if the authors probe the lysates from figure 1 for MMP2 and 9 to show this phenomenon in these cells.
RESPONSE: Thanks for the observation, in fact, the gelatin zymogram shown in Figure 6, was carried out with the culture medium obtained from cells exhibiting increasing percentages of ELC compared to the decreasing in fibroblasts, from 30 to 100 % of ELC.
Figure 8: Could the authors provide western blot or electron microscope data shown efficient isolation of microvesicles. If the authors probe for exosome markers such as TSG101 or CD63 or microvesicle markers such as CD40 then this would be sufficient.
RESPONSE: The technique used in obtaining the MVs was based in a commercial Kit (ExoQuick-TC™ Exosome Precipitation Solution; Cat# EXOTC50A-1), which was used to obtain exosomes and MVs. The purity of the MVs was characterized with immunoblots for CD63 and Hsp70. Thus in Figure 8A is now include a representative image from an immunoblot of the CD-63 protein in MVs obtained from ELC from an IPF patient.
The isolation of Microvesicles (MVs) section now is rephrased so:
2.10. Isolation of Microvesicles (MVs)
To assess whether MVs obtained from ELC induced the phenotypic change in human normal lung fibroblasts (passage 4), two different fibroblast cultures: one culture from normal human lung and one culture from idiopathic pulmonary fibrosis (IPF; passage 4), were treated for eight to ten days (~4 media changes) until ELC were observed (passage 4). The technique used in obtaining the MVs was based on a commercial Kit (ExoQuick-TC™ Exosome Precipitation Solution; Cat# EXOTC50A-1; System Biosciences), following the manufacturer's recommendations. Secreted MVs were recovered by differential centrifugation from conditioned media obtained from fibroblasts grown 24 h without serum. Briefly, 80 ml of collected media were centrifuged at 1500 xg for 10 min at 4°C, followed by centrifugation at 3,200 xg for 10 min at 4°C min, to remove dead cells and cell debris. The supernatants were recovered and centrifuged again at 20,000 xg at 4°C for two h to isolate the pellet with MVs [13, 14]. The purity of the MVs obtained was confirmed by immunoblotting with CD-63 specific antibodies. Subsequently, the expression of E-cadherin was assessed by Western blotting, and phase-contrast microscopy images were obtained to examine the morphologic change from fibroblast to ELC.
MVs were characterized with the antibodies to CD63 (cat. SBI #EXOAB-CD63A-1; rabbit anti-human) with goat anti-rabbit HRP secondary antibody; System Biosciences; 1:500 dilution), and Hsp70 (cat. EXOAB- Hsp 70A-1; System Biosciences; 1:500 dilution) with goat anti-rabbit HRP secondary antibody. Thus, in section 2.5. Immunoblotting. Was write:
MVs were characterized with the anti-human CD63 antibody (cat. SBI #EXOAB-CD63A-1; rabbit anti-human) with goat anti-rabbit HRP secondary antibody; System Biosciences; 1:500 dilution).
A representative immunoblot image to CD63 is now included in the new Figure 8B.
Legend to Figure 8 was rewritten:
Figure 8. Microvesicles (MVs) obtained from ELC induce the change to an epithelial-like phenotype in human normal lung fibroblasts. Representative Immunoblot image to the CD63 protein in MVs purified from ECL from an IPF patient (A). MVs effect on E-cadherin expression analyzed by western blotting: MVs purified from culture media obtained from human normal lung fibroblasts (NF; passage 4), IPF fibroblasts (IPF; passage 4), and ELC (passage 4). Total protein extract from A549 epithelial cell line was used as a positive control (B). Serial images of human lung fibroblasts incubated with MVs converted in ELC, observed with phase-contrast microscopy (original magnification x10): area with 100 % of normal lung fibroblasts (C); Image captured after 2 days exhibiting ~95 % fibroblasts and ~5 % ELC displaying a polygonal/cobblestone shaped, closely connected to each other and focally accumulated (D); Image obtained after 4 days exhibiting ~90 % fibroblasts and ~10 % of ELC (E); Image acquired after 6 days showing ~65 % fibroblasts and ~35% of ELC (F); Image took after 8 days displaying ~40 % fibroblasts and ~60 % of ELC (G); Image obtained after 10 days displaying ~20 % fibroblasts and ~80 % ELC (H); Area covered at ~100 % by ELC captured after 12 days of culture, where fibroblasts are no apparent (I).
Reference 14 was corrected.
Quackenbush, J.F.; Cassidy, P.B.; Pfeffer, LM; Boucher, K.M.; Hawkes, J.R.; Pfeffer, S.R.; Kopelovich, L.; Leachman, S.A. (2014) Isolation of Circulating MicroRNAs from Microvesicles Found in Human Plasma. In Molecular Diagnostics for Melanoma. Methods in Molecular Biology (Methods and Protocols); Editors Thurin M., Marincola F.; Humana Press, Totowa, NJ. Volume 1102, pp. 641-653. 2014;1102:641-53. doi: 10.1007/978-1-62703-727-3_34.
Table 3: The authors state miRNAs that are increased are shown in bold italics but this is not the case. It would be more informative for the authors to shown changes in the miRNA as fold differences. Besides, the authors describe miRNA in results section 3.7 that is not present in the data presented in table 3.
RESPONSE: Thanks for the observation, effectively is wrong the label of miRNAs. However, we prefer to modify Table 3 including the fold change for the eleven miRNA overexpressed in MVs derived from fibroblasts compared with ELC. Indicating in the text the only miRNA overexpressed in ELC vs. fibroblasts.
In Section 3.7. were changed the title and the text, including the Fold Changes to the miRNA overexpression in fibroblasts and ELC. Table 3 also was modified.
3.7. MVs from ELC Showed miRNAs Specific and Downregulated, but one Overexpressed Compared to Fibroblast
MVs are composed of lipids, proteins, and microRNAs (miRNAs). We examined the miRNA expression in the MVs from fibroblasts and ELC to determine whether some of these molecules are involved in MET. MicroRNA expression was evaluated in the MVs of two cell lines by quantitative RT-PCR. A total of 6 miRNAs were exclusively expressed in MVs from fibroblasts: hsa-miR-10a-3p, hsa-miR-125b-1-3p, hsa-miR-214-5p, hsa-miR-376a-5p, hsa-miR-380-3p, and hsa-miR-1471. Three miRNAs were exclusively expressed in MVs from ELC: hsa-miRNA-29b, a microRNA implicated in MET (see below), hsa-miR-96-5p and hsa-miR-935. Similarly, twelve miRNAs were differentially expressed between fibroblasts and ELC; eleven were overexpressed in Fibroblasts compared with ELC (Table 3), but one of these, hsa-miR-30b-5p, was overexpressed in ELC versus fibroblasts.
Table 3. miRNAs overexpressed in MVs derived from fibroblasts compared with those from ELC. Fold changes are indicated (FC). F: fibroblast; ELC: epithelial-like cell.
|
F vs ELC |
FC |
|
hsa-let-7a-5p |
3.1 |
|
hsa-let-7b-5p |
66.5 |
|
hsa-let-7c |
4.4 |
|
hsa-let-7d-5p |
19.7 |
|
hsa-let-7e-5p |
5.2 |
|
hsa-miR-21-3p |
8.1 |
|
hsa-miR-23a-3p |
3.0 |
|
hsa-miR-23b-3p |
4.9 |
|
hsa-miR-24-2-5p |
2.2 |
|
hsa-miR-26a-5p |
49.4 |
|
hsa-miR-27a-3p |
16.8 |
|
ELC vs F |
|
|
hsa-miR-30b-5p |
2.4 |
Results section 3.8 is difficult to understand and therefore it is difficult to review this section of the manuscript. Could the authors describe this data more clarity.
RESPONSE: It's a good point, this paragraph was rewritten.
Finally, once we showed the differences between the original fibroblasts and the ELC resulting from the spontaneous conversion, we proceeded to verify the phenotype of the fibroblasts that are routinely cultured in our laboratory. For this purpose, we stained the cells with antibodies against proteins expressed, specifically in fibroblasts (Thy-1 cell surface antigen, vimentin, type I collagen, PDGFRb), and with an antibody against pan-cytokeratin, a specific protein present in epithelial cells but not in fibroblast. As expected, the fibroblasts are positive for fibroblast-markers and negative for epithelial cells, corroborating that they are phenotypically indeed fibroblasts, as shown in Figure 9.
Additionally.
Spelling, grammar, and punctuation were revised across the entire manuscript; the changes are indicated in red script.

Round 2
Reviewer 1 Report
I believe the manuscript is significantly improved. Thank you for addressing all my comments. One last question regarding figure 4:
- Were the fibroblasts-rich culture plates obtained from different passages of the same cell line compared to the ELC-enriched ones? If so, please specify it in the figure caption.
Author Response
Reply to the Review Report (Reviewer 1)
Journal Title: Biomolecules
Article Title: ´´Mesenchymal–Epithelial Transition in Fibroblasts of Human Normal Lungs and Interstitial Lung Diseases ´´.
Thanks a lot to editors and reviewers for commenting on our manuscript.
Following the reviewer suggestions, we made several changes to the manuscript, which were written in red and shaded with yellow.
Below is our point-by-point response to the reviewer's comments and the actions taken to amend the manuscript, indicating the page where the change appears, as well as the new references as they appear in the new manuscript. We hope that the revised manuscript is acceptable to be published in the prominent journal Biomolecules
Author's Reply to the Review Report (Reviewer 1)
(x) I would not like to sign my review report
( ) I would like to sign my review report
English language and style
( ) Extensive editing of English language and style required
(x) Moderate English changes required
( ) English language and style are fine/minor spell check required
( ) I don't feel qualified to judge about the English language and style
Yes Can be improved Must be improved Not applicable
Does the introduction provide sufficient background and include all relevant references? (x) ( ) ( ) ( )
Is the research design appropriate? (x) ( ) ( ) ( )
Are the methods adequately described? (x) ( ) ( ) ( )
Are the results clearly presented? (x) ( ) ( ) ( )
Are the conclusions supported by the results? (x) ( ) ( ) ( )
Comments and Suggestions for Authors
I believe the manuscript is significantly improved. Thank you for addressing all my comments. One last question regarding figure 4:
- Were the fibroblasts-rich culture plates obtained from different passages of the same cell line compared to the ELC-enriched ones? If so, please specify it in the figure caption.
Submission Date
31 January 2021
Date of this review
7 Feb 2021 00:2
Reply to the Review Report (Reviewer 1)
I believe the manuscript is significantly improved. Thank you for addressing all my comments. One last question regarding figure 4:
- Were the fibroblasts-rich culture plates obtained from different passages of the same cell line compared to the ELC-enriched ones? If so, please specify it in the figure caption.
RESPONSE: Effectively, the fibroblasts-rich culture plates obtained from different passages of the same cell line were compared with the correspondent ELC-enriched ones.
The change is indicated with red text and shaded with yellow.
The new legend to figure 4 is now:
Page 10, lines 318-323.
Figure 4. E-cadherin was increased in ELC derived from human lung fibroblasts. The fibroblasts-rich culture plates obtained from different passages of the same cell line were compared with the correspondent ELC. The cells were fixed, labeled with an E-cadherin (CD324) antibody (PE-Cy5.5), and analyzed by cytofluorometry. Percentage of cells expressing E-cadherin in four different fibroblast and ELC lines (A): two were obtained from ILD patients (Line-1 and Line-2), and two from brain dead patients (Line-3 and Line-4). Representative scatter dot-plot images of flow cytometry for E-cadherin in fibroblasts and ELC (B). F: fibroblast; ELC: epithelial-like cell.

Reviewer 2 Report
The authors have addressed the experimental questions I raised in the first review. I am happy for the manuscript to be published after a few minor changes in the text
I believe the manuscript would benefit from the authors expanding the text in results section 3.6. The authors jump straight into there MVs work without introducing what MVs are and what are there functions. In addition it would help if the authors explained their rationale for looking at the role of MVs at the beginning of this section.
The authors should carefully review the new text in section 2.7. There are a few grammatical errors in this new section
Author Response
Reply to the Review Report (Reviewer 2)
Journal Title: Biomolecules
Article Title: ´´Mesenchymal–Epithelial Transition in Fibroblasts of Human Normal Lungs and Interstitial Lung Diseases ´´.
Thanks a lot to editors and reviewers for commenting on our manuscript.
Following the reviewer suggestions, we made several changes to the manuscript. Changes are written in red and shaded with yellow.
Below is our point-by-point response to the reviewer's comments and the actions taken to amend the manuscript, indicating the page where the change appears, as well as the new references as they appear in the new manuscript. We hope that the revised manuscript is acceptable to be published in the prominent journal Biomolecules.
Author's Reply to the Review Report (Reviewer 2)
Reviewer’s comments:
Open Review
(x) I would not like to sign my review report
( ) I would like to sign my review report
English language and style
( ) Extensive editing of English language and style required
( ) Moderate English changes required
( ) English language and style are fine/minor spell check required
(x) I don't feel qualified to judge about the English language and style
Yes Can be improved Must be improved Not applicable
Does the introduction provide sufficient background and include all relevant references?
(x) ( ) ( ) ( )
Is the research design appropriate?
(x) ( ) ( ) ( )
Are the methods adequately described?
(x) ( ) ( ) ( )
Are the results clearly presented?
(x) ( ) ( ) ( )
Are the conclusions supported by the results?
(x) ( ) ( ) ( )
Comments and Suggestions for Authors
The authors have addressed the experimental questions I raised in the first review. I am happy for the manuscript to be published after a few minor changes in the text
I believe the manuscript would benefit from the authors expanding the text in results section 3.6. The authors jump straight into there MVs work without introducing what MVs are and what are there functions. In addition it would help if the authors explained their rationale for looking at the role of MVs at the beginning of this section.
The authors should carefully review the new text in section 2.7. There are a few grammatical errors in this
new section
Submission Date
31 January 2021
Date of this review
16 Feb 2021 13:18:32
Reply to the Review Report (Reviewer 2)
I believe the manuscript would benefit from the authors expanding the text in results section 3.6. The authors jump straight into there MVs work without introducing what MVs are and what are there functions. In addition it would help if the authors explained their rationale for looking at the role of MVs at the beginning of this section.
RESPONSE: A brief description of the MVs characteristics and their functions is now included in section 3.6. The change is indicated with red text and shaded with yellow. Page 15, lines 408-415.
MVs are heterogeneous, membrane-bound sacs, shed from the surface of several cell types, having a size from 100 to 1000 nm, capable of encapsulating and transferring multiple forms of cargo, including proteins, mRNA, and miRNAs, siRNA, and plasmid DNA. MVs have relevant roles in transforming the extracellular environment, intercellular signaling, facilitating cell invasion through cell-independent matrix proteolysis, and modifying genetic expression and cell physiology in the proximal or distal recipient cells. MVs are found in multiple bodily fluids and tissues and could significantly impact future diagnostic and therapeutic strategies [36].
The authors should carefully review the new text in section 2.7. There are a few grammatical errors in this new section
RESPONSE: Thanks for the observation. The gelatin zymography method described in section 2.7 is now rewritten and included in the new version of the manuscript. The change is indicated with red text and shaded with yellow.
Page 4, lines 197-199, and Page 5, lines 200-202.
Gelatin zymography is a powerful and useful technique for measuring the relative amounts of active and inactive gelatinases (zymogens) in aqueous samples by measuring gelatin hydrolysis. The enzymes MMP-2 (gelatinase A) and MMP-9 (gelatinase B) are fractionated in SDS-PAGE gels, to be activated by denaturation and detected by including gelatin as a substrate in the gel. The gelatin hydrolysis produce two or three white bands, for each, zymogen or active forms, after staining with Coomassie blue [12].
Reply to the Review Report (Reviewer 2)
Journal Title: Biomolecules
Article Title: ´´Mesenchymal–Epithelial Transition in Fibroblasts of Human Normal Lungs and Interstitial Lung Diseases ´´.
Thanks a lot to editors and reviewers for commenting on our manuscript.
Following the reviewer suggestions, we made several changes to the manuscript. Changes are written in red and shaded with yellow.
Below is our point-by-point response to the reviewer's comments and the actions taken to amend the manuscript, indicating the page where the change appears, as well as the new references as they appear in the new manuscript. We hope that the revised manuscript is acceptable to be published in the prominent journal Biomolecules.
Author's Reply to the Review Report (Reviewer 2)
Reviewer’s comments:
Open Review
(x) I would not like to sign my review report
( ) I would like to sign my review report
English language and style
( ) Extensive editing of English language and style required
( ) Moderate English changes required
( ) English language and style are fine/minor spell check required
(x) I don't feel qualified to judge about the English language and style
Yes Can be improved Must be improved Not applicable
Does the introduction provide sufficient background and include all relevant references?
(x) ( ) ( ) ( )
Is the research design appropriate?
(x) ( ) ( ) ( )
Are the methods adequately described?
(x) ( ) ( ) ( )
Are the results clearly presented?
(x) ( ) ( ) ( )
Are the conclusions supported by the results?
(x) ( ) ( ) ( )
Comments and Suggestions for Authors
The authors have addressed the experimental questions I raised in the first review. I am happy for the manuscript to be published after a few minor changes in the text
I believe the manuscript would benefit from the authors expanding the text in results section 3.6. The authors jump straight into there MVs work without introducing what MVs are and what are there functions. In addition it would help if the authors explained their rationale for looking at the role of MVs at the beginning of this section.
The authors should carefully review the new text in section 2.7. There are a few grammatical errors in this
new section
Submission Date
31 January 2021
Date of this review
16 Feb 2021 13:18:32
Reply to the Review Report (Reviewer 2)
I believe the manuscript would benefit from the authors expanding the text in results section 3.6. The authors jump straight into there MVs work without introducing what MVs are and what are there functions. In addition it would help if the authors explained their rationale for looking at the role of MVs at the beginning of this section.
RESPONSE: A brief description of the MVs characteristics and their functions is now included in section 3.6. The change is indicated with red text and shaded with yellow. Page 15, lines 408-415.
MVs are heterogeneous, membrane-bound sacs, shed from the surface of several cell types, having a size from 100 to 1000 nm, capable of encapsulating and transferring multiple forms of cargo, including proteins, mRNA, and miRNAs, siRNA, and plasmid DNA. MVs have relevant roles in transforming the extracellular environment, intercellular signaling, facilitating cell invasion through cell-independent matrix proteolysis, and modifying genetic expression and cell physiology in the proximal or distal recipient cells. MVs are found in multiple bodily fluids and tissues and could significantly impact future diagnostic and therapeutic strategies [36].
The authors should carefully review the new text in section 2.7. There are a few grammatical errors in this new section
RESPONSE: Thanks for the observation. The gelatin zymography method described in section 2.7 is now rewritten and included in the new version of the manuscript. The change is indicated with red text and shaded with yellow.
Page 4, lines 197-199, and Page 5, lines 200-202.
Gelatin zymography is a powerful and useful technique for measuring the relative amounts of active and inactive gelatinases (zymogens) in aqueous samples by measuring gelatin hydrolysis. The enzymes MMP-2 (gelatinase A) and MMP-9 (gelatinase B) are fractionated in SDS-PAGE gels, to be activated by denaturation and detected by including gelatin as a substrate in the gel. The gelatin hydrolysis produce two or three white bands, for each, zymogen or active forms, after staining with Coomassie blue [12].
Reply to the Review Report (Reviewer 2)
Journal Title: Biomolecules
Article Title: ´´Mesenchymal–Epithelial Transition in Fibroblasts of Human Normal Lungs and Interstitial Lung Diseases ´´.
Thanks a lot to editors and reviewers for commenting on our manuscript.
Following the reviewer suggestions, we made several changes to the manuscript. Changes are written in red and shaded with yellow.
Below is our point-by-point response to the reviewer's comments and the actions taken to amend the manuscript, indicating the page where the change appears, as well as the new references as they appear in the new manuscript. We hope that the revised manuscript is acceptable to be published in the prominent journal Biomolecules.
Author's Reply to the Review Report (Reviewer 2)
Reviewer’s comments:
Open Review
(x) I would not like to sign my review report
( ) I would like to sign my review report
English language and style
( ) Extensive editing of English language and style required
( ) Moderate English changes required
( ) English language and style are fine/minor spell check required
(x) I don't feel qualified to judge about the English language and style
Yes Can be improved Must be improved Not applicable
Does the introduction provide sufficient background and include all relevant references?
(x) ( ) ( ) ( )
Is the research design appropriate?
(x) ( ) ( ) ( )
Are the methods adequately described?
(x) ( ) ( ) ( )
Are the results clearly presented?
(x) ( ) ( ) ( )
Are the conclusions supported by the results?
(x) ( ) ( ) ( )
Comments and Suggestions for Authors
The authors have addressed the experimental questions I raised in the first review. I am happy for the manuscript to be published after a few minor changes in the text
I believe the manuscript would benefit from the authors expanding the text in results section 3.6. The authors jump straight into there MVs work without introducing what MVs are and what are there functions. In addition it would help if the authors explained their rationale for looking at the role of MVs at the beginning of this section.
The authors should carefully review the new text in section 2.7. There are a few grammatical errors in this
new section
Submission Date
31 January 2021
Date of this review
16 Feb 2021 13:18:32
Reply to the Review Report (Reviewer 2)
I believe the manuscript would benefit from the authors expanding the text in results section 3.6. The authors jump straight into there MVs work without introducing what MVs are and what are there functions. In addition it would help if the authors explained their rationale for looking at the role of MVs at the beginning of this section.
RESPONSE: A brief description of the MVs characteristics and their functions is now included in section 3.6. The change is indicated with red text and shaded with yellow. Page 15, lines 408-415.
MVs are heterogeneous, membrane-bound sacs, shed from the surface of several cell types, having a size from 100 to 1000 nm, capable of encapsulating and transferring multiple forms of cargo, including proteins, mRNA, and miRNAs, siRNA, and plasmid DNA. MVs have relevant roles in transforming the extracellular environment, intercellular signaling, facilitating cell invasion through cell-independent matrix proteolysis, and modifying genetic expression and cell physiology in the proximal or distal recipient cells. MVs are found in multiple bodily fluids and tissues and could significantly impact future diagnostic and therapeutic strategies [36].
The authors should carefully review the new text in section 2.7. There are a few grammatical errors in this new section
RESPONSE: Thanks for the observation. The gelatin zymography method described in section 2.7 is now rewritten and included in the new version of the manuscript. The change is indicated with red text and shaded with yellow.
Page 4, lines 197-199, and Page 5, lines 200-202.
Gelatin zymography is a powerful and useful technique for measuring the relative amounts of active and inactive gelatinases (zymogens) in aqueous samples by measuring gelatin hydrolysis. The enzymes MMP-2 (gelatinase A) and MMP-9 (gelatinase B) are fractionated in SDS-PAGE gels, to be activated by denaturation and detected by including gelatin as a substrate in the gel. The gelatin hydrolysis produce two or three white bands, for each, zymogen or active forms, after staining with Coomassie blue [12].
Reply to the Review Report (Reviewer 2)
Journal Title: Biomolecules
Article Title: ´´Mesenchymal–Epithelial Transition in Fibroblasts of Human Normal Lungs and Interstitial Lung Diseases ´´.
Thanks a lot to editors and reviewers for commenting on our manuscript.
Following the reviewer suggestions, we made several changes to the manuscript. Changes are written in red and shaded with yellow.
Below is our point-by-point response to the reviewer's comments and the actions taken to amend the manuscript, indicating the page where the change appears, as well as the new references as they appear in the new manuscript. We hope that the revised manuscript is acceptable to be published in the prominent journal Biomolecules.
Author's Reply to the Review Report (Reviewer 2)
Reviewer’s comments:
Open Review
(x) I would not like to sign my review report
( ) I would like to sign my review report
English language and style
( ) Extensive editing of English language and style required
( ) Moderate English changes required
( ) English language and style are fine/minor spell check required
(x) I don't feel qualified to judge about the English language and style
Yes Can be improved Must be improved Not applicable
Does the introduction provide sufficient background and include all relevant references?
(x) ( ) ( ) ( )
Is the research design appropriate?
(x) ( ) ( ) ( )
Are the methods adequately described?
(x) ( ) ( ) ( )
Are the results clearly presented?
(x) ( ) ( ) ( )
Are the conclusions supported by the results?
(x) ( ) ( ) ( )
Comments and Suggestions for Authors
The authors have addressed the experimental questions I raised in the first review. I am happy for the manuscript to be published after a few minor changes in the text
I believe the manuscript would benefit from the authors expanding the text in results section 3.6. The authors jump straight into there MVs work without introducing what MVs are and what are there functions. In addition it would help if the authors explained their rationale for looking at the role of MVs at the beginning of this section.
The authors should carefully review the new text in section 2.7. There are a few grammatical errors in this
new section
Submission Date
31 January 2021
Date of this review
16 Feb 2021 13:18:32
Reply to the Review Report (Reviewer 2)
I believe the manuscript would benefit from the authors expanding the text in results section 3.6. The authors jump straight into there MVs work without introducing what MVs are and what are there functions. In addition it would help if the authors explained their rationale for looking at the role of MVs at the beginning of this section.
RESPONSE: A brief description of the MVs characteristics and their functions is now included in section 3.6. The change is indicated with red text and shaded with yellow. Page 15, lines 408-415.
MVs are heterogeneous, membrane-bound sacs, shed from the surface of several cell types, having a size from 100 to 1000 nm, capable of encapsulating and transferring multiple forms of cargo, including proteins, mRNA, and miRNAs, siRNA, and plasmid DNA. MVs have relevant roles in transforming the extracellular environment, intercellular signaling, facilitating cell invasion through cell-independent matrix proteolysis, and modifying genetic expression and cell physiology in the proximal or distal recipient cells. MVs are found in multiple bodily fluids and tissues and could significantly impact future diagnostic and therapeutic strategies [36].
The authors should carefully review the new text in section 2.7. There are a few grammatical errors in this new section
RESPONSE: Thanks for the observation. The gelatin zymography method described in section 2.7 is now rewritten and included in the new version of the manuscript. The change is indicated with red text and shaded with yellow.
Page 4, lines 197-199, and Page 5, lines 200-202.
Gelatin zymography is a powerful and useful technique for measuring the relative amounts of active and inactive gelatinases (zymogens) in aqueous samples by measuring gelatin hydrolysis. The enzymes MMP-2 (gelatinase A) and MMP-9 (gelatinase B) are fractionated in SDS-PAGE gels, to be activated by denaturation and detected by including gelatin as a substrate in the gel. The gelatin hydrolysis produce two or three white bands, for each, zymogen or active forms, after staining with Coomassie blue [12].
Reply to the Review Report (Reviewer 2)
Journal Title: Biomolecules
Article Title: ´´Mesenchymal–Epithelial Transition in Fibroblasts of Human Normal Lungs and Interstitial Lung Diseases ´´.
Thanks a lot to editors and reviewers for commenting on our manuscript.
Following the reviewer suggestions, we made several changes to the manuscript. Changes are written in red and shaded with yellow.
Below is our point-by-point response to the reviewer's comments and the actions taken to amend the manuscript, indicating the page where the change appears, as well as the new references as they appear in the new manuscript. We hope that the revised manuscript is acceptable to be published in the prominent journal Biomolecules.
Author's Reply to the Review Report (Reviewer 2)
Reviewer’s comments:
Open Review
(x) I would not like to sign my review report
( ) I would like to sign my review report
English language and style
( ) Extensive editing of English language and style required
( ) Moderate English changes required
( ) English language and style are fine/minor spell check required
(x) I don't feel qualified to judge about the English language and style
Yes Can be improved Must be improved Not applicable
Does the introduction provide sufficient background and include all relevant references?
(x) ( ) ( ) ( )
Is the research design appropriate?
(x) ( ) ( ) ( )
Are the methods adequately described?
(x) ( ) ( ) ( )
Are the results clearly presented?
(x) ( ) ( ) ( )
Are the conclusions supported by the results?
(x) ( ) ( ) ( )
Comments and Suggestions for Authors
The authors have addressed the experimental questions I raised in the first review. I am happy for the manuscript to be published after a few minor changes in the text
I believe the manuscript would benefit from the authors expanding the text in results section 3.6. The authors jump straight into there MVs work without introducing what MVs are and what are there functions. In addition it would help if the authors explained their rationale for looking at the role of MVs at the beginning of this section.
The authors should carefully review the new text in section 2.7. There are a few grammatical errors in this
new section
Submission Date
31 January 2021
Date of this review
16 Feb 2021 13:18:32
Reply to the Review Report (Reviewer 2)
I believe the manuscript would benefit from the authors expanding the text in results section 3.6. The authors jump straight into there MVs work without introducing what MVs are and what are there functions. In addition it would help if the authors explained their rationale for looking at the role of MVs at the beginning of this section.
RESPONSE: A brief description of the MVs characteristics and their functions is now included in section 3.6. The change is indicated with red text and shaded with yellow. Page 15, lines 408-415.
MVs are heterogeneous, membrane-bound sacs, shed from the surface of several cell types, having a size from 100 to 1000 nm, capable of encapsulating and transferring multiple forms of cargo, including proteins, mRNA, and miRNAs, siRNA, and plasmid DNA. MVs have relevant roles in transforming the extracellular environment, intercellular signaling, facilitating cell invasion through cell-independent matrix proteolysis, and modifying genetic expression and cell physiology in the proximal or distal recipient cells. MVs are found in multiple bodily fluids and tissues and could significantly impact future diagnostic and therapeutic strategies [36].
The authors should carefully review the new text in section 2.7. There are a few grammatical errors in this new section
RESPONSE: Thanks for the observation. The gelatin zymography method described in section 2.7 is now rewritten and included in the new version of the manuscript. The change is indicated with red text and shaded with yellow.
Page 4, lines 197-199, and Page 5, lines 200-202.
Gelatin zymography is a powerful and useful technique for measuring the relative amounts of active and inactive gelatinases (zymogens) in aqueous samples by measuring gelatin hydrolysis. The enzymes MMP-2 (gelatinase A) and MMP-9 (gelatinase B) are fractionated in SDS-PAGE gels, to be activated by denaturation and detected by including gelatin as a substrate in the gel. The gelatin hydrolysis produce two or three white bands, for each, zymogen or active forms, after staining with Coomassie blue [12].
Reply to the Review Report (Reviewer 2)
Journal Title: Biomolecules
Article Title: ´´Mesenchymal–Epithelial Transition in Fibroblasts of Human Normal Lungs and Interstitial Lung Diseases ´´.
Thanks a lot to editors and reviewers for commenting on our manuscript.
Following the reviewer suggestions, we made several changes to the manuscript. Changes are written in red and shaded with yellow.
Below is our point-by-point response to the reviewer's comments and the actions taken to amend the manuscript, indicating the page where the change appears, as well as the new references as they appear in the new manuscript. We hope that the revised manuscript is acceptable to be published in the prominent journal Biomolecules.
Author's Reply to the Review Report (Reviewer 2)
Reviewer’s comments:
Open Review
(x) I would not like to sign my review report
( ) I would like to sign my review report
English language and style
( ) Extensive editing of English language and style required
( ) Moderate English changes required
( ) English language and style are fine/minor spell check required
(x) I don't feel qualified to judge about the English language and style
Yes Can be improved Must be improved Not applicable
Does the introduction provide sufficient background and include all relevant references?
(x) ( ) ( ) ( )
Is the research design appropriate?
(x) ( ) ( ) ( )
Are the methods adequately described?
(x) ( ) ( ) ( )
Are the results clearly presented?
(x) ( ) ( ) ( )
Are the conclusions supported by the results?
(x) ( ) ( ) ( )
Comments and Suggestions for Authors
The authors have addressed the experimental questions I raised in the first review. I am happy for the manuscript to be published after a few minor changes in the text
I believe the manuscript would benefit from the authors expanding the text in results section 3.6. The authors jump straight into there MVs work without introducing what MVs are and what are there functions. In addition it would help if the authors explained their rationale for looking at the role of MVs at the beginning of this section.
The authors should carefully review the new text in section 2.7. There are a few grammatical errors in this
new section
Submission Date
31 January 2021
Date of this review
16 Feb 2021 13:18:32
Reply to the Review Report (Reviewer 2)
I believe the manuscript would benefit from the authors expanding the text in results section 3.6. The authors jump straight into there MVs work without introducing what MVs are and what are there functions. In addition it would help if the authors explained their rationale for looking at the role of MVs at the beginning of this section.
RESPONSE: A brief description of the MVs characteristics and their functions is now included in section 3.6. The change is indicated with red text and shaded with yellow. Page 15, lines 408-415.
MVs are heterogeneous, membrane-bound sacs, shed from the surface of several cell types, having a size from 100 to 1000 nm, capable of encapsulating and transferring multiple forms of cargo, including proteins, mRNA, and miRNAs, siRNA, and plasmid DNA. MVs have relevant roles in transforming the extracellular environment, intercellular signaling, facilitating cell invasion through cell-independent matrix proteolysis, and modifying genetic expression and cell physiology in the proximal or distal recipient cells. MVs are found in multiple bodily fluids and tissues and could significantly impact future diagnostic and therapeutic strategies [36].
The authors should carefully review the new text in section 2.7. There are a few grammatical errors in this new section
RESPONSE: Thanks for the observation. The gelatin zymography method described in section 2.7 is now rewritten and included in the new version of the manuscript. The change is indicated with red text and shaded with yellow.
Page 4, lines 197-199, and Page 5, lines 200-202.
Gelatin zymography is a powerful and useful technique for measuring the relative amounts of active and inactive gelatinases (zymogens) in aqueous samples by measuring gelatin hydrolysis. The enzymes MMP-2 (gelatinase A) and MMP-9 (gelatinase B) are fractionated in SDS-PAGE gels, to be activated by denaturation and detected by including gelatin as a substrate in the gel. The gelatin hydrolysis produce two or three white bands, for each, zymogen or active forms, after staining with Coomassie blue [12].
Reply to the Review Report (Reviewer 2)
Journal Title: Biomolecules
Article Title: ´´Mesenchymal–Epithelial Transition in Fibroblasts of Human Normal Lungs and Interstitial Lung Diseases ´´.
Thanks a lot to editors and reviewers for commenting on our manuscript.
Following the reviewer suggestions, we made several changes to the manuscript. Changes are written in red and shaded with yellow.
Below is our point-by-point response to the reviewer's comments and the actions taken to amend the manuscript, indicating the page where the change appears, as well as the new references as they appear in the new manuscript. We hope that the revised manuscript is acceptable to be published in the prominent journal Biomolecules.
Author's Reply to the Review Report (Reviewer 2)
Reviewer’s comments:
Open Review
(x) I would not like to sign my review report
( ) I would like to sign my review report
English language and style
( ) Extensive editing of English language and style required
( ) Moderate English changes required
( ) English language and style are fine/minor spell check required
(x) I don't feel qualified to judge about the English language and style
Yes Can be improved Must be improved Not applicable
Does the introduction provide sufficient background and include all relevant references?
(x) ( ) ( ) ( )
Is the research design appropriate?
(x) ( ) ( ) ( )
Are the methods adequately described?
(x) ( ) ( ) ( )
Are the results clearly presented?
(x) ( ) ( ) ( )
Are the conclusions supported by the results?
(x) ( ) ( ) ( )
Comments and Suggestions for Authors
The authors have addressed the experimental questions I raised in the first review. I am happy for the manuscript to be published after a few minor changes in the text
I believe the manuscript would benefit from the authors expanding the text in results section 3.6. The authors jump straight into there MVs work without introducing what MVs are and what are there functions. In addition it would help if the authors explained their rationale for looking at the role of MVs at the beginning of this section.
The authors should carefully review the new text in section 2.7. There are a few grammatical errors in this
new section
Submission Date
31 January 2021
Date of this review
16 Feb 2021 13:18:32
Reply to the Review Report (Reviewer 2)
I believe the manuscript would benefit from the authors expanding the text in results section 3.6. The authors jump straight into there MVs work without introducing what MVs are and what are there functions. In addition it would help if the authors explained their rationale for looking at the role of MVs at the beginning of this section.
RESPONSE: A brief description of the MVs characteristics and their functions is now included in section 3.6. The change is indicated with red text and shaded with yellow. Page 15, lines 408-415.
MVs are heterogeneous, membrane-bound sacs, shed from the surface of several cell types, having a size from 100 to 1000 nm, capable of encapsulating and transferring multiple forms of cargo, including proteins, mRNA, and miRNAs, siRNA, and plasmid DNA. MVs have relevant roles in transforming the extracellular environment, intercellular signaling, facilitating cell invasion through cell-independent matrix proteolysis, and modifying genetic expression and cell physiology in the proximal or distal recipient cells. MVs are found in multiple bodily fluids and tissues and could significantly impact future diagnostic and therapeutic strategies [36].
The authors should carefully review the new text in section 2.7. There are a few grammatical errors in this new section
RESPONSE: Thanks for the observation. The gelatin zymography method described in section 2.7 is now rewritten and included in the new version of the manuscript. The change is indicated with red text and shaded with yellow.
Page 4, lines 197-199, and Page 5, lines 200-202.
Gelatin zymography is a powerful and useful technique for measuring the relative amounts of active and inactive gelatinases (zymogens) in aqueous samples by measuring gelatin hydrolysis. The enzymes MMP-2 (gelatinase A) and MMP-9 (gelatinase B) are fractionated in SDS-PAGE gels, to be activated by denaturation and detected by including gelatin as a substrate in the gel. The gelatin hydrolysis produce two or three white bands, for each, zymogen or active forms, after staining with Coomassie blue [12].
Reply to the Review Report (Reviewer 2)
Journal Title: Biomolecules
Article Title: ´´Mesenchymal–Epithelial Transition in Fibroblasts of Human Normal Lungs and Interstitial Lung Diseases ´´.
Thanks a lot to editors and reviewers for commenting on our manuscript.
Following the reviewer suggestions, we made several changes to the manuscript. Changes are written in red and shaded with yellow.
Below is our point-by-point response to the reviewer's comments and the actions taken to amend the manuscript, indicating the page where the change appears, as well as the new references as they appear in the new manuscript. We hope that the revised manuscript is acceptable to be published in the prominent journal Biomolecules.
Author's Reply to the Review Report (Reviewer 2)
Reviewer’s comments:
Open Review
(x) I would not like to sign my review report
( ) I would like to sign my review report
English language and style
( ) Extensive editing of English language and style required
( ) Moderate English changes required
( ) English language and style are fine/minor spell check required
(x) I don't feel qualified to judge about the English language and style
Yes Can be improved Must be improved Not applicable
Does the introduction provide sufficient background and include all relevant references?
(x) ( ) ( ) ( )
Is the research design appropriate?
(x) ( ) ( ) ( )
Are the methods adequately described?
(x) ( ) ( ) ( )
Are the results clearly presented?
(x) ( ) ( ) ( )
Are the conclusions supported by the results?
(x) ( ) ( ) ( )
Comments and Suggestions for Authors
The authors have addressed the experimental questions I raised in the first review. I am happy for the manuscript to be published after a few minor changes in the text
I believe the manuscript would benefit from the authors expanding the text in results section 3.6. The authors jump straight into there MVs work without introducing what MVs are and what are there functions. In addition it would help if the authors explained their rationale for looking at the role of MVs at the beginning of this section.
The authors should carefully review the new text in section 2.7. There are a few grammatical errors in this
new section
Submission Date
31 January 2021
Date of this review
16 Feb 2021 13:18:32
Reply to the Review Report (Reviewer 2)
I believe the manuscript would benefit from the authors expanding the text in results section 3.6. The authors jump straight into there MVs work without introducing what MVs are and what are there functions. In addition it would help if the authors explained their rationale for looking at the role of MVs at the beginning of this section.
RESPONSE: A brief description of the MVs characteristics and their functions is now included in section 3.6. The change is indicated with red text and shaded with yellow. Page 15, lines 408-415.
MVs are heterogeneous, membrane-bound sacs, shed from the surface of several cell types, having a size from 100 to 1000 nm, capable of encapsulating and transferring multiple forms of cargo, including proteins, mRNA, and miRNAs, siRNA, and plasmid DNA. MVs have relevant roles in transforming the extracellular environment, intercellular signaling, facilitating cell invasion through cell-independent matrix proteolysis, and modifying genetic expression and cell physiology in the proximal or distal recipient cells. MVs are found in multiple bodily fluids and tissues and could significantly impact future diagnostic and therapeutic strategies [36].
The authors should carefully review the new text in section 2.7. There are a few grammatical errors in this new section
RESPONSE: Thanks for the observation. The gelatin zymography method described in section 2.7 is now rewritten and included in the new version of the manuscript. The change is indicated with red text and shaded with yellow.
Page 4, lines 197-199, and Page 5, lines 200-202.
Gelatin zymography is a powerful and useful technique for measuring the relative amounts of active and inactive gelatinases (zymogens) in aqueous samples by measuring gelatin hydrolysis. The enzymes MMP-2 (gelatinase A) and MMP-9 (gelatinase B) are fractionated in SDS-PAGE gels, to be activated by denaturation and detected by including gelatin as a substrate in the gel. The gelatin hydrolysis produce two or three white bands, for each, zymogen or active forms, after staining with Coomassie blue [12].
Reply to the Review Report (Reviewer 2)
Journal Title: Biomolecules
Article Title: ´´Mesenchymal–Epithelial Transition in Fibroblasts of Human Normal Lungs and Interstitial Lung Diseases ´´.
Thanks a lot to editors and reviewers for commenting on our manuscript.
Following the reviewer suggestions, we made several changes to the manuscript. Changes are written in red and shaded with yellow.
Below is our point-by-point response to the reviewer's comments and the actions taken to amend the manuscript, indicating the page where the change appears, as well as the new references as they appear in the new manuscript. We hope that the revised manuscript is acceptable to be published in the prominent journal Biomolecules.
Author's Reply to the Review Report (Reviewer 2)
Reviewer’s comments:
Open Review
(x) I would not like to sign my review report
( ) I would like to sign my review report
English language and style
( ) Extensive editing of English language and style required
( ) Moderate English changes required
( ) English language and style are fine/minor spell check required
(x) I don't feel qualified to judge about the English language and style
Yes Can be improved Must be improved Not applicable
Does the introduction provide sufficient background and include all relevant references?
(x) ( ) ( ) ( )
Is the research design appropriate?
(x) ( ) ( ) ( )
Are the methods adequately described?
(x) ( ) ( ) ( )
Are the results clearly presented?
(x) ( ) ( ) ( )
Are the conclusions supported by the results?
(x) ( ) ( ) ( )
Comments and Suggestions for Authors
The authors have addressed the experimental questions I raised in the first review. I am happy for the manuscript to be published after a few minor changes in the text
I believe the manuscript would benefit from the authors expanding the text in results section 3.6. The authors jump straight into there MVs work without introducing what MVs are and what are there functions. In addition it would help if the authors explained their rationale for looking at the role of MVs at the beginning of this section.
The authors should carefully review the new text in section 2.7. There are a few grammatical errors in this
new section
Submission Date
31 January 2021
Date of this review
16 Feb 2021 13:18:32
Reply to the Review Report (Reviewer 2)
I believe the manuscript would benefit from the authors expanding the text in results section 3.6. The authors jump straight into there MVs work without introducing what MVs are and what are there functions. In addition it would help if the authors explained their rationale for looking at the role of MVs at the beginning of this section.
RESPONSE: A brief description of the MVs characteristics and their functions is now included in section 3.6. The change is indicated with red text and shaded with yellow. Page 15, lines 408-415.
MVs are heterogeneous, membrane-bound sacs, shed from the surface of several cell types, having a size from 100 to 1000 nm, capable of encapsulating and transferring multiple forms of cargo, including proteins, mRNA, and miRNAs, siRNA, and plasmid DNA. MVs have relevant roles in transforming the extracellular environment, intercellular signaling, facilitating cell invasion through cell-independent matrix proteolysis, and modifying genetic expression and cell physiology in the proximal or distal recipient cells. MVs are found in multiple bodily fluids and tissues and could significantly impact future diagnostic and therapeutic strategies [36].
The authors should carefully review the new text in section 2.7. There are a few grammatical errors in this new section
RESPONSE: Thanks for the observation. The gelatin zymography method described in section 2.7 is now rewritten and included in the new version of the manuscript. The change is indicated with red text and shaded with yellow.
Page 4, lines 197-199, and Page 5, lines 200-202.
Gelatin zymography is a powerful and useful technique for measuring the relative amounts of active and inactive gelatinases (zymogens) in aqueous samples by measuring gelatin hydrolysis. The enzymes MMP-2 (gelatinase A) and MMP-9 (gelatinase B) are fractionated in SDS-PAGE gels, to be activated by denaturation and detected by including gelatin as a substrate in the gel. The gelatin hydrolysis produce two or three white bands, for each, zymogen or active forms, after staining with Coomassie blue [12].
Reply to the Review Report (Reviewer 2)
Journal Title: Biomolecules
Article Title: ´´Mesenchymal–Epithelial Transition in Fibroblasts of Human Normal Lungs and Interstitial Lung Diseases ´´.
Thanks a lot to editors and reviewers for commenting on our manuscript.
Following the reviewer suggestions, we made several changes to the manuscript. Changes are written in red and shaded with yellow.
Below is our point-by-point response to the reviewer's comments and the actions taken to amend the manuscript, indicating the page where the change appears, as well as the new references as they appear in the new manuscript. We hope that the revised manuscript is acceptable to be published in the prominent journal Biomolecules.
Author's Reply to the Review Report (Reviewer 2)
Reviewer’s comments:
Open Review
(x) I would not like to sign my review report
( ) I would like to sign my review report
English language and style
( ) Extensive editing of English language and style required
( ) Moderate English changes required
( ) English language and style are fine/minor spell check required
(x) I don't feel qualified to judge about the English language and style
Yes Can be improved Must be improved Not applicable
Does the introduction provide sufficient background and include all relevant references?
(x) ( ) ( ) ( )
Is the research design appropriate?
(x) ( ) ( ) ( )
Are the methods adequately described?
(x) ( ) ( ) ( )
Are the results clearly presented?
(x) ( ) ( ) ( )
Are the conclusions supported by the results?
(x) ( ) ( ) ( )
Comments and Suggestions for Authors
The authors have addressed the experimental questions I raised in the first review. I am happy for the manuscript to be published after a few minor changes in the text
I believe the manuscript would benefit from the authors expanding the text in results section 3.6. The authors jump straight into there MVs work without introducing what MVs are and what are there functions. In addition it would help if the authors explained their rationale for looking at the role of MVs at the beginning of this section.
The authors should carefully review the new text in section 2.7. There are a few grammatical errors in this
new section
Submission Date
31 January 2021
Date of this review
16 Feb 2021 13:18:32
Reply to the Review Report (Reviewer 2)
I believe the manuscript would benefit from the authors expanding the text in results section 3.6. The authors jump straight into there MVs work without introducing what MVs are and what are there functions. In addition it would help if the authors explained their rationale for looking at the role of MVs at the beginning of this section.
RESPONSE: A brief description of the MVs characteristics and their functions is now included in section 3.6. The change is indicated with red text and shaded with yellow. Page 15, lines 408-415.
MVs are heterogeneous, membrane-bound sacs, shed from the surface of several cell types, having a size from 100 to 1000 nm, capable of encapsulating and transferring multiple forms of cargo, including proteins, mRNA, and miRNAs, siRNA, and plasmid DNA. MVs have relevant roles in transforming the extracellular environment, intercellular signaling, facilitating cell invasion through cell-independent matrix proteolysis, and modifying genetic expression and cell physiology in the proximal or distal recipient cells. MVs are found in multiple bodily fluids and tissues and could significantly impact future diagnostic and therapeutic strategies [36].
The authors should carefully review the new text in section 2.7. There are a few grammatical errors in this new section
RESPONSE: Thanks for the observation. The gelatin zymography method described in section 2.7 is now rewritten and included in the new version of the manuscript. The change is indicated with red text and shaded with yellow.
Page 4, lines 197-199, and Page 5, lines 200-202.
Gelatin zymography is a powerful and useful technique for measuring the relative amounts of active and inactive gelatinases (zymogens) in aqueous samples by measuring gelatin hydrolysis. The enzymes MMP-2 (gelatinase A) and MMP-9 (gelatinase B) are fractionated in SDS-PAGE gels, to be activated by denaturation and detected by including gelatin as a substrate in the gel. The gelatin hydrolysis produce two or three white bands, for each, zymogen or active forms, after staining with Coomassie blue [12].
Reply to the Review Report (Reviewer 2)
Journal Title: Biomolecules
Article Title: ´´Mesenchymal–Epithelial Transition in Fibroblasts of Human Normal Lungs and Interstitial Lung Diseases ´´.
Thanks a lot to editors and reviewers for commenting on our manuscript.
Following the reviewer suggestions, we made several changes to the manuscript. Changes are written in red and shaded with yellow.
Below is our point-by-point response to the reviewer's comments and the actions taken to amend the manuscript, indicating the page where the change appears, as well as the new references as they appear in the new manuscript. We hope that the revised manuscript is acceptable to be published in the prominent journal Biomolecules.
Author's Reply to the Review Report (Reviewer 2)
Reviewer’s comments:
Open Review
(x) I would not like to sign my review report
( ) I would like to sign my review report
English language and style
( ) Extensive editing of English language and style required
( ) Moderate English changes required
( ) English language and style are fine/minor spell check required
(x) I don't feel qualified to judge about the English language and style
Yes Can be improved Must be improved Not applicable
Does the introduction provide sufficient background and include all relevant references?
(x) ( ) ( ) ( )
Is the research design appropriate?
(x) ( ) ( ) ( )
Are the methods adequately described?
(x) ( ) ( ) ( )
Are the results clearly presented?
(x) ( ) ( ) ( )
Are the conclusions supported by the results?
(x) ( ) ( ) ( )
Comments and Suggestions for Authors
The authors have addressed the experimental questions I raised in the first review. I am happy for the manuscript to be published after a few minor changes in the text
I believe the manuscript would benefit from the authors expanding the text in results section 3.6. The authors jump straight into there MVs work without introducing what MVs are and what are there functions. In addition it would help if the authors explained their rationale for looking at the role of MVs at the beginning of this section.
The authors should carefully review the new text in section 2.7. There are a few grammatical errors in this
new section
Submission Date
31 January 2021
Date of this review
16 Feb 2021 13:18:32
Reply to the Review Report (Reviewer 2)
I believe the manuscript would benefit from the authors expanding the text in results section 3.6. The authors jump straight into there MVs work without introducing what MVs are and what are there functions. In addition it would help if the authors explained their rationale for looking at the role of MVs at the beginning of this section.
RESPONSE: A brief description of the MVs characteristics and their functions is now included in section 3.6. The change is indicated with red text and shaded with yellow. Page 15, lines 408-415.
MVs are heterogeneous, membrane-bound sacs, shed from the surface of several cell types, having a size from 100 to 1000 nm, capable of encapsulating and transferring multiple forms of cargo, including proteins, mRNA, and miRNAs, siRNA, and plasmid DNA. MVs have relevant roles in transforming the extracellular environment, intercellular signaling, facilitating cell invasion through cell-independent matrix proteolysis, and modifying genetic expression and cell physiology in the proximal or distal recipient cells. MVs are found in multiple bodily fluids and tissues and could significantly impact future diagnostic and therapeutic strategies [36].
The authors should carefully review the new text in section 2.7. There are a few grammatical errors in this new section
RESPONSE: Thanks for the observation. The gelatin zymography method described in section 2.7 is now rewritten and included in the new version of the manuscript. The change is indicated with red text and shaded with yellow.
Page 4, lines 197-199, and Page 5, lines 200-202.
Gelatin zymography is a powerful and useful technique for measuring the relative amounts of active and inactive gelatinases (zymogens) in aqueous samples by measuring gelatin hydrolysis. The enzymes MMP-2 (gelatinase A) and MMP-9 (gelatinase B) are fractionated in SDS-PAGE gels, to be activated by denaturation and detected by including gelatin as a substrate in the gel. The gelatin hydrolysis produce two or three white bands, for each, zymogen or active forms, after staining with Coomassie blue [12].
Reply to the Review Report (Reviewer 2)
Journal Title: Biomolecules
Article Title: ´´Mesenchymal–Epithelial Transition in Fibroblasts of Human Normal Lungs and Interstitial Lung Diseases ´´.
Thanks a lot to editors and reviewers for commenting on our manuscript.
Following the reviewer suggestions, we made several changes to the manuscript. Changes are written in red and shaded with yellow.
Below is our point-by-point response to the reviewer's comments and the actions taken to amend the manuscript, indicating the page where the change appears, as well as the new references as they appear in the new manuscript. We hope that the revised manuscript is acceptable to be published in the prominent journal Biomolecules.
Author's Reply to the Review Report (Reviewer 2)
Reviewer’s comments:
Open Review
(x) I would not like to sign my review report
( ) I would like to sign my review report
English language and style
( ) Extensive editing of English language and style required
( ) Moderate English changes required
( ) English language and style are fine/minor spell check required
(x) I don't feel qualified to judge about the English language and style
Yes Can be improved Must be improved Not applicable
Does the introduction provide sufficient background and include all relevant references?
(x) ( ) ( ) ( )
Is the research design appropriate?
(x) ( ) ( ) ( )
Are the methods adequately described?
(x) ( ) ( ) ( )
Are the results clearly presented?
(x) ( ) ( ) ( )
Are the conclusions supported by the results?
(x) ( ) ( ) ( )
Comments and Suggestions for Authors
The authors have addressed the experimental questions I raised in the first review. I am happy for the manuscript to be published after a few minor changes in the text
I believe the manuscript would benefit from the authors expanding the text in results section 3.6. The authors jump straight into there MVs work without introducing what MVs are and what are there functions. In addition it would help if the authors explained their rationale for looking at the role of MVs at the beginning of this section.
The authors should carefully review the new text in section 2.7. There are a few grammatical errors in this
new section
Submission Date
31 January 2021
Date of this review
16 Feb 2021 13:18:32
Reply to the Review Report (Reviewer 2)
I believe the manuscript would benefit from the authors expanding the text in results section 3.6. The authors jump straight into there MVs work without introducing what MVs are and what are there functions. In addition it would help if the authors explained their rationale for looking at the role of MVs at the beginning of this section.
RESPONSE: A brief description of the MVs characteristics and their functions is now included in section 3.6. The change is indicated with red text and shaded with yellow. Page 15, lines 408-415.
MVs are heterogeneous, membrane-bound sacs, shed from the surface of several cell types, having a size from 100 to 1000 nm, capable of encapsulating and transferring multiple forms of cargo, including proteins, mRNA, and miRNAs, siRNA, and plasmid DNA. MVs have relevant roles in transforming the extracellular environment, intercellular signaling, facilitating cell invasion through cell-independent matrix proteolysis, and modifying genetic expression and cell physiology in the proximal or distal recipient cells. MVs are found in multiple bodily fluids and tissues and could significantly impact future diagnostic and therapeutic strategies [36].
The authors should carefully review the new text in section 2.7. There are a few grammatical errors in this new section
RESPONSE: Thanks for the observation. The gelatin zymography method described in section 2.7 is now rewritten and included in the new version of the manuscript. The change is indicated with red text and shaded with yellow.
Page 4, lines 197-199, and Page 5, lines 200-202.
Gelatin zymography is a powerful and useful technique for measuring the relative amounts of active and inactive gelatinases (zymogens) in aqueous samples by measuring gelatin hydrolysis. The enzymes MMP-2 (gelatinase A) and MMP-9 (gelatinase B) are fractionated in SDS-PAGE gels, to be activated by denaturation and detected by including gelatin as a substrate in the gel. The gelatin hydrolysis produce two or three white bands, for each, zymogen or active forms, after staining with Coomassie blue [12].
Reply to the Review Report (Reviewer 2)
Journal Title: Biomolecules
Article Title: ´´Mesenchymal–Epithelial Transition in Fibroblasts of Human Normal Lungs and Interstitial Lung Diseases ´´.
Thanks a lot to editors and reviewers for commenting on our manuscript.
Following the reviewer suggestions, we made several changes to the manuscript. Changes are written in red and shaded with yellow.
Below is our point-by-point response to the reviewer's comments and the actions taken to amend the manuscript, indicating the page where the change appears, as well as the new references as they appear in the new manuscript. We hope that the revised manuscript is acceptable to be published in the prominent journal Biomolecules.
Author's Reply to the Review Report (Reviewer 2)
Reviewer’s comments:
Open Review
(x) I would not like to sign my review report
( ) I would like to sign my review report
English language and style
( ) Extensive editing of English language and style required
( ) Moderate English changes required
( ) English language and style are fine/minor spell check required
(x) I don't feel qualified to judge about the English language and style
Yes Can be improved Must be improved Not applicable
Does the introduction provide sufficient background and include all relevant references?
(x) ( ) ( ) ( )
Is the research design appropriate?
(x) ( ) ( ) ( )
Are the methods adequately described?
(x) ( ) ( ) ( )
Are the results clearly presented?
(x) ( ) ( ) ( )
Are the conclusions supported by the results?
(x) ( ) ( ) ( )
Comments and Suggestions for Authors
The authors have addressed the experimental questions I raised in the first review. I am happy for the manuscript to be published after a few minor changes in the text
I believe the manuscript would benefit from the authors expanding the text in results section 3.6. The authors jump straight into there MVs work without introducing what MVs are and what are there functions. In addition it would help if the authors explained their rationale for looking at the role of MVs at the beginning of this section.
The authors should carefully review the new text in section 2.7. There are a few grammatical errors in this
new section
Submission Date
31 January 2021
Date of this review
16 Feb 2021 13:18:32
Reply to the Review Report (Reviewer 2)
I believe the manuscript would benefit from the authors expanding the text in results section 3.6. The authors jump straight into there MVs work without introducing what MVs are and what are there functions. In addition it would help if the authors explained their rationale for looking at the role of MVs at the beginning of this section.
RESPONSE: A brief description of the MVs characteristics and their functions is now included in section 3.6. The change is indicated with red text and shaded with yellow. Page 15, lines 408-415.
MVs are heterogeneous, membrane-bound sacs, shed from the surface of several cell types, having a size from 100 to 1000 nm, capable of encapsulating and transferring multiple forms of cargo, including proteins, mRNA, and miRNAs, siRNA, and plasmid DNA. MVs have relevant roles in transforming the extracellular environment, intercellular signaling, facilitating cell invasion through cell-independent matrix proteolysis, and modifying genetic expression and cell physiology in the proximal or distal recipient cells. MVs are found in multiple bodily fluids and tissues and could significantly impact future diagnostic and therapeutic strategies [36].
The authors should carefully review the new text in section 2.7. There are a few grammatical errors in this new section
RESPONSE: Thanks for the observation. The gelatin zymography method described in section 2.7 is now rewritten and included in the new version of the manuscript. The change is indicated with red text and shaded with yellow.
Page 4, lines 197-199, and Page 5, lines 200-202.
Gelatin zymography is a powerful and useful technique for measuring the relative amounts of active and inactive gelatinases (zymogens) in aqueous samples by measuring gelatin hydrolysis. The enzymes MMP-2 (gelatinase A) and MMP-9 (gelatinase B) are fractionated in SDS-PAGE gels, to be activated by denaturation and detected by including gelatin as a substrate in the gel. The gelatin hydrolysis produce two or three white bands, for each, zymogen or active forms, after staining with Coomassie blue [12].
Reply to the Review Report (Reviewer 2)
Journal Title: Biomolecules
Article Title: ´´Mesenchymal–Epithelial Transition in Fibroblasts of Human Normal Lungs and Interstitial Lung Diseases ´´.
Thanks a lot to editors and reviewers for commenting on our manuscript.
Following the reviewer suggestions, we made several changes to the manuscript. Changes are written in red and shaded with yellow.
Below is our point-by-point response to the reviewer's comments and the actions taken to amend the manuscript, indicating the page where the change appears, as well as the new references as they appear in the new manuscript. We hope that the revised manuscript is acceptable to be published in the prominent journal Biomolecules.
Author's Reply to the Review Report (Reviewer 2)
Reviewer’s comments:
Open Review
(x) I would not like to sign my review report
( ) I would like to sign my review report
English language and style
( ) Extensive editing of English language and style required
( ) Moderate English changes required
( ) English language and style are fine/minor spell check required
(x) I don't feel qualified to judge about the English language and style
Yes Can be improved Must be improved Not applicable
Does the introduction provide sufficient background and include all relevant references?
(x) ( ) ( ) ( )
Is the research design appropriate?
(x) ( ) ( ) ( )
Are the methods adequately described?
(x) ( ) ( ) ( )
Are the results clearly presented?
(x) ( ) ( ) ( )
Are the conclusions supported by the results?
(x) ( ) ( ) ( )
Comments and Suggestions for Authors
The authors have addressed the experimental questions I raised in the first review. I am happy for the manuscript to be published after a few minor changes in the text
I believe the manuscript would benefit from the authors expanding the text in results section 3.6. The authors jump straight into there MVs work without introducing what MVs are and what are there functions. In addition it would help if the authors explained their rationale for looking at the role of MVs at the beginning of this section.
The authors should carefully review the new text in section 2.7. There are a few grammatical errors in this
new section
Submission Date
31 January 2021
Date of this review
16 Feb 2021 13:18:32
Reply to the Review Report (Reviewer 2)
I believe the manuscript would benefit from the authors expanding the text in results section 3.6. The authors jump straight into there MVs work without introducing what MVs are and what are there functions. In addition it would help if the authors explained their rationale for looking at the role of MVs at the beginning of this section.
RESPONSE: A brief description of the MVs characteristics and their functions is now included in section 3.6. The change is indicated with red text and shaded with yellow. Page 15, lines 408-415.
MVs are heterogeneous, membrane-bound sacs, shed from the surface of several cell types, having a size from 100 to 1000 nm, capable of encapsulating and transferring multiple forms of cargo, including proteins, mRNA, and miRNAs, siRNA, and plasmid DNA. MVs have relevant roles in transforming the extracellular environment, intercellular signaling, facilitating cell invasion through cell-independent matrix proteolysis, and modifying genetic expression and cell physiology in the proximal or distal recipient cells. MVs are found in multiple bodily fluids and tissues and could significantly impact future diagnostic and therapeutic strategies [36].
The authors should carefully review the new text in section 2.7. There are a few grammatical errors in this new section
RESPONSE: Thanks for the observation. The gelatin zymography method described in section 2.7 is now rewritten and included in the new version of the manuscript. The change is indicated with red text and shaded with yellow.
Page 4, lines 197-199, and Page 5, lines 200-202.
Gelatin zymography is a powerful and useful technique for measuring the relative amounts of active and inactive gelatinases (zymogens) in aqueous samples by measuring gelatin hydrolysis. The enzymes MMP-2 (gelatinase A) and MMP-9 (gelatinase B) are fractionated in SDS-PAGE gels, to be activated by denaturation and detected by including gelatin as a substrate in the gel. The gelatin hydrolysis produce two or three white bands, for each, zymogen or active forms, after staining with Coomassie blue [12].
Reply to the Review Report (Reviewer 2)
Journal Title: Biomolecules
Article Title: ´´Mesenchymal–Epithelial Transition in Fibroblasts of Human Normal Lungs and Interstitial Lung Diseases ´´.
Thanks a lot to editors and reviewers for commenting on our manuscript.
Following the reviewer suggestions, we made several changes to the manuscript. Changes are written in red and shaded with yellow.
Below is our point-by-point response to the reviewer's comments and the actions taken to amend the manuscript, indicating the page where the change appears, as well as the new references as they appear in the new manuscript. We hope that the revised manuscript is acceptable to be published in the prominent journal Biomolecules.
Author's Reply to the Review Report (Reviewer 2)
Reviewer’s comments:
Open Review
(x) I would not like to sign my review report
( ) I would like to sign my review report
English language and style
( ) Extensive editing of English language and style required
( ) Moderate English changes required
( ) English language and style are fine/minor spell check required
(x) I don't feel qualified to judge about the English language and style
Yes Can be improved Must be improved Not applicable
Does the introduction provide sufficient background and include all relevant references?
(x) ( ) ( ) ( )
Is the research design appropriate?
(x) ( ) ( ) ( )
Are the methods adequately described?
(x) ( ) ( ) ( )
Are the results clearly presented?
(x) ( ) ( ) ( )
Are the conclusions supported by the results?
(x) ( ) ( ) ( )
Comments and Suggestions for Authors
The authors have addressed the experimental questions I raised in the first review. I am happy for the manuscript to be published after a few minor changes in the text
I believe the manuscript would benefit from the authors expanding the text in results section 3.6. The authors jump straight into there MVs work without introducing what MVs are and what are there functions. In addition it would help if the authors explained their rationale for looking at the role of MVs at the beginning of this section.
The authors should carefully review the new text in section 2.7. There are a few grammatical errors in this
new section
Submission Date
31 January 2021
Date of this review
16 Feb 2021 13:18:32
Reply to the Review Report (Reviewer 2)
I believe the manuscript would benefit from the authors expanding the text in results section 3.6. The authors jump straight into there MVs work without introducing what MVs are and what are there functions. In addition it would help if the authors explained their rationale for looking at the role of MVs at the beginning of this section.
RESPONSE: A brief description of the MVs characteristics and their functions is now included in section 3.6. The change is indicated with red text and shaded with yellow. Page 15, lines 408-415.
MVs are heterogeneous, membrane-bound sacs, shed from the surface of several cell types, having a size from 100 to 1000 nm, capable of encapsulating and transferring multiple forms of cargo, including proteins, mRNA, and miRNAs, siRNA, and plasmid DNA. MVs have relevant roles in transforming the extracellular environment, intercellular signaling, facilitating cell invasion through cell-independent matrix proteolysis, and modifying genetic expression and cell physiology in the proximal or distal recipient cells. MVs are found in multiple bodily fluids and tissues and could significantly impact future diagnostic and therapeutic strategies [36].
The authors should carefully review the new text in section 2.7. There are a few grammatical errors in this new section
RESPONSE: Thanks for the observation. The gelatin zymography method described in section 2.7 is now rewritten and included in the new version of the manuscript. The change is indicated with red text and shaded with yellow.
Page 4, lines 197-199, and Page 5, lines 200-202.
Gelatin zymography is a powerful and useful technique for measuring the relative amounts of active and inactive gelatinases (zymogens) in aqueous samples by measuring gelatin hydrolysis. The enzymes MMP-2 (gelatinase A) and MMP-9 (gelatinase B) are fractionated in SDS-PAGE gels, to be activated by denaturation and detected by including gelatin as a substrate in the gel. The gelatin hydrolysis produce two or three white bands, for each, zymogen or active forms, after staining with Coomassie blue [12].
Reply to the Review Report (Reviewer 2)
Journal Title: Biomolecules
Article Title: ´´Mesenchymal–Epithelial Transition in Fibroblasts of Human Normal Lungs and Interstitial Lung Diseases ´´.
Thanks a lot to editors and reviewers for commenting on our manuscript.
Following the reviewer suggestions, we made several changes to the manuscript. Changes are written in red and shaded with yellow.
Below is our point-by-point response to the reviewer's comments and the actions taken to amend the manuscript, indicating the page where the change appears, as well as the new references as they appear in the new manuscript. We hope that the revised manuscript is acceptable to be published in the prominent journal Biomolecules.
Author's Reply to the Review Report (Reviewer 2)
Reviewer’s comments:
Open Review
(x) I would not like to sign my review report
( ) I would like to sign my review report
English language and style
( ) Extensive editing of English language and style required
( ) Moderate English changes required
( ) English language and style are fine/minor spell check required
(x) I don't feel qualified to judge about the English language and style
Yes Can be improved Must be improved Not applicable
Does the introduction provide sufficient background and include all relevant references?
(x) ( ) ( ) ( )
Is the research design appropriate?
(x) ( ) ( ) ( )
Are the methods adequately described?
(x) ( ) ( ) ( )
Are the results clearly presented?
(x) ( ) ( ) ( )
Are the conclusions supported by the results?
(x) ( ) ( ) ( )
Comments and Suggestions for Authors
The authors have addressed the experimental questions I raised in the first review. I am happy for the manuscript to be published after a few minor changes in the text
I believe the manuscript would benefit from the authors expanding the text in results section 3.6. The authors jump straight into there MVs work without introducing what MVs are and what are there functions. In addition it would help if the authors explained their rationale for looking at the role of MVs at the beginning of this section.
The authors should carefully review the new text in section 2.7. There are a few grammatical errors in this
new section
Submission Date
31 January 2021
Date of this review
16 Feb 2021 13:18:32
Reply to the Review Report (Reviewer 2)
I believe the manuscript would benefit from the authors expanding the text in results section 3.6. The authors jump straight into there MVs work without introducing what MVs are and what are there functions. In addition it would help if the authors explained their rationale for looking at the role of MVs at the beginning of this section.
RESPONSE: A brief description of the MVs characteristics and their functions is now included in section 3.6. The change is indicated with red text and shaded with yellow. Page 15, lines 408-415.
MVs are heterogeneous, membrane-bound sacs, shed from the surface of several cell types, having a size from 100 to 1000 nm, capable of encapsulating and transferring multiple forms of cargo, including proteins, mRNA, and miRNAs, siRNA, and plasmid DNA. MVs have relevant roles in transforming the extracellular environment, intercellular signaling, facilitating cell invasion through cell-independent matrix proteolysis, and modifying genetic expression and cell physiology in the proximal or distal recipient cells. MVs are found in multiple bodily fluids and tissues and could significantly impact future diagnostic and therapeutic strategies [36].
The authors should carefully review the new text in section 2.7. There are a few grammatical errors in this new section
RESPONSE: Thanks for the observation. The gelatin zymography method described in section 2.7 is now rewritten and included in the new version of the manuscript. The change is indicated with red text and shaded with yellow.
Page 4, lines 197-199, and Page 5, lines 200-202.
Gelatin zymography is a powerful and useful technique for measuring the relative amounts of active and inactive gelatinases (zymogens) in aqueous samples by measuring gelatin hydrolysis. The enzymes MMP-2 (gelatinase A) and MMP-9 (gelatinase B) are fractionated in SDS-PAGE gels, to be activated by denaturation and detected by including gelatin as a substrate in the gel. The gelatin hydrolysis produce two or three white bands, for each, zymogen or active forms, after staining with Coomassie blue [12].
Reply to the Review Report (Reviewer 2)
Journal Title: Biomolecules
Article Title: ´´Mesenchymal–Epithelial Transition in Fibroblasts of Human Normal Lungs and Interstitial Lung Diseases ´´.
Thanks a lot to editors and reviewers for commenting on our manuscript.
Following the reviewer suggestions, we made several changes to the manuscript. Changes are written in red and shaded with yellow.
Below is our point-by-point response to the reviewer's comments and the actions taken to amend the manuscript, indicating the page where the change appears, as well as the new references as they appear in the new manuscript. We hope that the revised manuscript is acceptable to be published in the prominent journal Biomolecules.
Author's Reply to the Review Report (Reviewer 2)
Reviewer’s comments:
Open Review
(x) I would not like to sign my review report
( ) I would like to sign my review report
English language and style
( ) Extensive editing of English language and style required
( ) Moderate English changes required
( ) English language and style are fine/minor spell check required
(x) I don't feel qualified to judge about the English language and style
Yes Can be improved Must be improved Not applicable
Does the introduction provide sufficient background and include all relevant references?
(x) ( ) ( ) ( )
Is the research design appropriate?
(x) ( ) ( ) ( )
Are the methods adequately described?
(x) ( ) ( ) ( )
Are the results clearly presented?
(x) ( ) ( ) ( )
Are the conclusions supported by the results?
(x) ( ) ( ) ( )
Comments and Suggestions for Authors
The authors have addressed the experimental questions I raised in the first review. I am happy for the manuscript to be published after a few minor changes in the text
I believe the manuscript would benefit from the authors expanding the text in results section 3.6. The authors jump straight into there MVs work without introducing what MVs are and what are there functions. In addition it would help if the authors explained their rationale for looking at the role of MVs at the beginning of this section.
The authors should carefully review the new text in section 2.7. There are a few grammatical errors in this
new section
Submission Date
31 January 2021
Date of this review
16 Feb 2021 13:18:32
Reply to the Review Report (Reviewer 2)
I believe the manuscript would benefit from the authors expanding the text in results section 3.6. The authors jump straight into there MVs work without introducing what MVs are and what are there functions. In addition it would help if the authors explained their rationale for looking at the role of MVs at the beginning of this section.
RESPONSE: A brief description of the MVs characteristics and their functions is now included in section 3.6. The change is indicated with red text and shaded with yellow. Page 15, lines 408-415.
MVs are heterogeneous, membrane-bound sacs, shed from the surface of several cell types, having a size from 100 to 1000 nm, capable of encapsulating and transferring multiple forms of cargo, including proteins, mRNA, and miRNAs, siRNA, and plasmid DNA. MVs have relevant roles in transforming the extracellular environment, intercellular signaling, facilitating cell invasion through cell-independent matrix proteolysis, and modifying genetic expression and cell physiology in the proximal or distal recipient cells. MVs are found in multiple bodily fluids and tissues and could significantly impact future diagnostic and therapeutic strategies [36].
The authors should carefully review the new text in section 2.7. There are a few grammatical errors in this new section
RESPONSE: Thanks for the observation. The gelatin zymography method described in section 2.7 is now rewritten and included in the new version of the manuscript. The change is indicated with red text and shaded with yellow.
Page 4, lines 197-199, and Page 5, lines 200-202.
Gelatin zymography is a powerful and useful technique for measuring the relative amounts of active and inactive gelatinases (zymogens) in aqueous samples by measuring gelatin hydrolysis. The enzymes MMP-2 (gelatinase A) and MMP-9 (gelatinase B) are fractionated in SDS-PAGE gels, to be activated by denaturation and detected by including gelatin as a substrate in the gel. The gelatin hydrolysis produce two or three white bands, for each, zymogen or active forms, after staining with Coomassie blue [12].
Reply to the Review Report (Reviewer 2)
Journal Title: Biomolecules
Article Title: ´´Mesenchymal–Epithelial Transition in Fibroblasts of Human Normal Lungs and Interstitial Lung Diseases ´´.
Thanks a lot to editors and reviewers for commenting on our manuscript.
Following the reviewer suggestions, we made several changes to the manuscript. Changes are written in red and shaded with yellow.
Below is our point-by-point response to the reviewer's comments and the actions taken to amend the manuscript, indicating the page where the change appears, as well as the new references as they appear in the new manuscript. We hope that the revised manuscript is acceptable to be published in the prominent journal Biomolecules.
Author's Reply to the Review Report (Reviewer 2)
Reviewer’s comments:
Open Review
(x) I would not like to sign my review report
( ) I would like to sign my review report
English language and style
( ) Extensive editing of English language and style required
( ) Moderate English changes required
( ) English language and style are fine/minor spell check required
(x) I don't feel qualified to judge about the English language and style
Yes Can be improved Must be improved Not applicable
Does the introduction provide sufficient background and include all relevant references?
(x) ( ) ( ) ( )
Is the research design appropriate?
(x) ( ) ( ) ( )
Are the methods adequately described?
(x) ( ) ( ) ( )
Are the results clearly presented?
(x) ( ) ( ) ( )
Are the conclusions supported by the results?
(x) ( ) ( ) ( )
Comments and Suggestions for Authors
The authors have addressed the experimental questions I raised in the first review. I am happy for the manuscript to be published after a few minor changes in the text
I believe the manuscript would benefit from the authors expanding the text in results section 3.6. The authors jump straight into there MVs work without introducing what MVs are and what are there functions. In addition it would help if the authors explained their rationale for looking at the role of MVs at the beginning of this section.
The authors should carefully review the new text in section 2.7. There are a few grammatical errors in this
new section
Submission Date
31 January 2021
Date of this review
16 Feb 2021 13:18:32
Reply to the Review Report (Reviewer 2)
I believe the manuscript would benefit from the authors expanding the text in results section 3.6. The authors jump straight into there MVs work without introducing what MVs are and what are there functions. In addition it would help if the authors explained their rationale for looking at the role of MVs at the beginning of this section.
RESPONSE: A brief description of the MVs characteristics and their functions is now included in section 3.6. The change is indicated with red text and shaded with yellow. Page 15, lines 408-415.
MVs are heterogeneous, membrane-bound sacs, shed from the surface of several cell types, having a size from 100 to 1000 nm, capable of encapsulating and transferring multiple forms of cargo, including proteins, mRNA, and miRNAs, siRNA, and plasmid DNA. MVs have relevant roles in transforming the extracellular environment, intercellular signaling, facilitating cell invasion through cell-independent matrix proteolysis, and modifying genetic expression and cell physiology in the proximal or distal recipient cells. MVs are found in multiple bodily fluids and tissues and could significantly impact future diagnostic and therapeutic strategies [36].
The authors should carefully review the new text in section 2.7. There are a few grammatical errors in this new section
RESPONSE: Thanks for the observation. The gelatin zymography method described in section 2.7 is now rewritten and included in the new version of the manuscript. The change is indicated with red text and shaded with yellow.
Page 4, lines 197-199, and Page 5, lines 200-202.
Gelatin zymography is a powerful and useful technique for measuring the relative amounts of active and inactive gelatinases (zymogens) in aqueous samples by measuring gelatin hydrolysis. The enzymes MMP-2 (gelatinase A) and MMP-9 (gelatinase B) are fractionated in SDS-PAGE gels, to be activated by denaturation and detected by including gelatin as a substrate in the gel. The gelatin hydrolysis produce two or three white bands, for each, zymogen or active forms, after staining with Coomassie blue [12].
Reply to the Review Report (Reviewer 2)
Journal Title: Biomolecules
Article Title: ´´Mesenchymal–Epithelial Transition in Fibroblasts of Human Normal Lungs and Interstitial Lung Diseases ´´.
Thanks a lot to editors and reviewers for commenting on our manuscript.
Following the reviewer suggestions, we made several changes to the manuscript. Changes are written in red and shaded with yellow.
Below is our point-by-point response to the reviewer's comments and the actions taken to amend the manuscript, indicating the page where the change appears, as well as the new references as they appear in the new manuscript. We hope that the revised manuscript is acceptable to be published in the prominent journal Biomolecules.
Author's Reply to the Review Report (Reviewer 2)
Reviewer’s comments:
Open Review
(x) I would not like to sign my review report
( ) I would like to sign my review report
English language and style
( ) Extensive editing of English language and style required
( ) Moderate English changes required
( ) English language and style are fine/minor spell check required
(x) I don't feel qualified to judge about the English language and style
Yes Can be improved Must be improved Not applicable
Does the introduction provide sufficient background and include all relevant references?
(x) ( ) ( ) ( )
Is the research design appropriate?
(x) ( ) ( ) ( )
Are the methods adequately described?
(x) ( ) ( ) ( )
Are the results clearly presented?
(x) ( ) ( ) ( )
Are the conclusions supported by the results?
(x) ( ) ( ) ( )
Comments and Suggestions for Authors
The authors have addressed the experimental questions I raised in the first review. I am happy for the manuscript to be published after a few minor changes in the text
I believe the manuscript would benefit from the authors expanding the text in results section 3.6. The authors jump straight into there MVs work without introducing what MVs are and what are there functions. In addition it would help if the authors explained their rationale for looking at the role of MVs at the beginning of this section.
The authors should carefully review the new text in section 2.7. There are a few grammatical errors in this
new section
Submission Date
31 January 2021
Date of this review
16 Feb 2021 13:18:32
Reply to the Review Report (Reviewer 2)
I believe the manuscript would benefit from the authors expanding the text in results section 3.6. The authors jump straight into there MVs work without introducing what MVs are and what are there functions. In addition it would help if the authors explained their rationale for looking at the role of MVs at the beginning of this section.
RESPONSE: A brief description of the MVs characteristics and their functions is now included in section 3.6. The change is indicated with red text and shaded with yellow. Page 15, lines 408-415.
MVs are heterogeneous, membrane-bound sacs, shed from the surface of several cell types, having a size from 100 to 1000 nm, capable of encapsulating and transferring multiple forms of cargo, including proteins, mRNA, and miRNAs, siRNA, and plasmid DNA. MVs have relevant roles in transforming the extracellular environment, intercellular signaling, facilitating cell invasion through cell-independent matrix proteolysis, and modifying genetic expression and cell physiology in the proximal or distal recipient cells. MVs are found in multiple bodily fluids and tissues and could significantly impact future diagnostic and therapeutic strategies [36].
The authors should carefully review the new text in section 2.7. There are a few grammatical errors in this new section
RESPONSE: Thanks for the observation. The gelatin zymography method described in section 2.7 is now rewritten and included in the new version of the manuscript. The change is indicated with red text and shaded with yellow.
Page 4, lines 197-199, and Page 5, lines 200-202.
Gelatin zymography is a powerful and useful technique for measuring the relative amounts of active and inactive gelatinases (zymogens) in aqueous samples by measuring gelatin hydrolysis. The enzymes MMP-2 (gelatinase A) and MMP-9 (gelatinase B) are fractionated in SDS-PAGE gels, to be activated by denaturation and detected by including gelatin as a substrate in the gel. The gelatin hydrolysis produce two or three white bands, for each, zymogen or active forms, after staining with Coomassie blue [12].
Reply to the Review Report (Reviewer 2)
Journal Title: Biomolecules
Article Title: ´´Mesenchymal–Epithelial Transition in Fibroblasts of Human Normal Lungs and Interstitial Lung Diseases ´´.
Thanks a lot to editors and reviewers for commenting on our manuscript.
Following the reviewer suggestions, we made several changes to the manuscript. Changes are written in red and shaded with yellow.
Below is our point-by-point response to the reviewer's comments and the actions taken to amend the manuscript, indicating the page where the change appears, as well as the new references as they appear in the new manuscript. We hope that the revised manuscript is acceptable to be published in the prominent journal Biomolecules.
Author's Reply to the Review Report (Reviewer 2)
Reviewer’s comments:
Open Review
(x) I would not like to sign my review report
( ) I would like to sign my review report
English language and style
( ) Extensive editing of English language and style required
( ) Moderate English changes required
( ) English language and style are fine/minor spell check required
(x) I don't feel qualified to judge about the English language and style
Yes Can be improved Must be improved Not applicable
Does the introduction provide sufficient background and include all relevant references?
(x) ( ) ( ) ( )
Is the research design appropriate?
(x) ( ) ( ) ( )
Are the methods adequately described?
(x) ( ) ( ) ( )
Are the results clearly presented?
(x) ( ) ( ) ( )
Are the conclusions supported by the results?
(x) ( ) ( ) ( )
Comments and Suggestions for Authors
The authors have addressed the experimental questions I raised in the first review. I am happy for the manuscript to be published after a few minor changes in the text
I believe the manuscript would benefit from the authors expanding the text in results section 3.6. The authors jump straight into there MVs work without introducing what MVs are and what are there functions. In addition it would help if the authors explained their rationale for looking at the role of MVs at the beginning of this section.
The authors should carefully review the new text in section 2.7. There are a few grammatical errors in this
new section
Submission Date
31 January 2021
Date of this review
16 Feb 2021 13:18:32
Reply to the Review Report (Reviewer 2)
I believe the manuscript would benefit from the authors expanding the text in results section 3.6. The authors jump straight into there MVs work without introducing what MVs are and what are there functions. In addition it would help if the authors explained their rationale for looking at the role of MVs at the beginning of this section.
RESPONSE: A brief description of the MVs characteristics and their functions is now included in section 3.6. The change is indicated with red text and shaded with yellow. Page 15, lines 408-415.
MVs are heterogeneous, membrane-bound sacs, shed from the surface of several cell types, having a size from 100 to 1000 nm, capable of encapsulating and transferring multiple forms of cargo, including proteins, mRNA, and miRNAs, siRNA, and plasmid DNA. MVs have relevant roles in transforming the extracellular environment, intercellular signaling, facilitating cell invasion through cell-independent matrix proteolysis, and modifying genetic expression and cell physiology in the proximal or distal recipient cells. MVs are found in multiple bodily fluids and tissues and could significantly impact future diagnostic and therapeutic strategies [36].
The authors should carefully review the new text in section 2.7. There are a few grammatical errors in this new section
RESPONSE: Thanks for the observation. The gelatin zymography method described in section 2.7 is now rewritten and included in the new version of the manuscript. The change is indicated with red text and shaded with yellow.
Page 4, lines 197-199, and Page 5, lines 200-202.
Gelatin zymography is a powerful and useful technique for measuring the relative amounts of active and inactive gelatinases (zymogens) in aqueous samples by measuring gelatin hydrolysis. The enzymes MMP-2 (gelatinase A) and MMP-9 (gelatinase B) are fractionated in SDS-PAGE gels, to be activated by denaturation and detected by including gelatin as a substrate in the gel. The gelatin hydrolysis produce two or three white bands, for each, zymogen or active forms, after staining with Coomassie blue [12].
Reply to the Review Report (Reviewer 2)
Journal Title: Biomolecules
Article Title: ´´Mesenchymal–Epithelial Transition in Fibroblasts of Human Normal Lungs and Interstitial Lung Diseases ´´.
Thanks a lot to editors and reviewers for commenting on our manuscript.
Following the reviewer suggestions, we made several changes to the manuscript. Changes are written in red and shaded with yellow.
Below is our point-by-point response to the reviewer's comments and the actions taken to amend the manuscript, indicating the page where the change appears, as well as the new references as they appear in the new manuscript. We hope that the revised manuscript is acceptable to be published in the prominent journal Biomolecules.
Author's Reply to the Review Report (Reviewer 2)
Reviewer’s comments:
Open Review
(x) I would not like to sign my review report
( ) I would like to sign my review report
English language and style
( ) Extensive editing of English language and style required
( ) Moderate English changes required
( ) English language and style are fine/minor spell check required
(x) I don't feel qualified to judge about the English language and style
Yes Can be improved Must be improved Not applicable
Does the introduction provide sufficient background and include all relevant references?
(x) ( ) ( ) ( )
Is the research design appropriate?
(x) ( ) ( ) ( )
Are the methods adequately described?
(x) ( ) ( ) ( )
Are the results clearly presented?
(x) ( ) ( ) ( )
Are the conclusions supported by the results?
(x) ( ) ( ) ( )
Comments and Suggestions for Authors
The authors have addressed the experimental questions I raised in the first review. I am happy for the manuscript to be published after a few minor changes in the text
I believe the manuscript would benefit from the authors expanding the text in results section 3.6. The authors jump straight into there MVs work without introducing what MVs are and what are there functions. In addition it would help if the authors explained their rationale for looking at the role of MVs at the beginning of this section.
The authors should carefully review the new text in section 2.7. There are a few grammatical errors in this
new section
Submission Date
31 January 2021
Date of this review
16 Feb 2021 13:18:32
Reply to the Review Report (Reviewer 2)
I believe the manuscript would benefit from the authors expanding the text in results section 3.6. The authors jump straight into there MVs work without introducing what MVs are and what are there functions. In addition it would help if the authors explained their rationale for looking at the role of MVs at the beginning of this section.
RESPONSE: A brief description of the MVs characteristics and their functions is now included in section 3.6. The change is indicated with red text and shaded with yellow. Page 15, lines 408-415.
MVs are heterogeneous, membrane-bound sacs, shed from the surface of several cell types, having a size from 100 to 1000 nm, capable of encapsulating and transferring multiple forms of cargo, including proteins, mRNA, and miRNAs, siRNA, and plasmid DNA. MVs have relevant roles in transforming the extracellular environment, intercellular signaling, facilitating cell invasion through cell-independent matrix proteolysis, and modifying genetic expression and cell physiology in the proximal or distal recipient cells. MVs are found in multiple bodily fluids and tissues and could significantly impact future diagnostic and therapeutic strategies [36].
The authors should carefully review the new text in section 2.7. There are a few grammatical errors in this new section
RESPONSE: Thanks for the observation. The gelatin zymography method described in section 2.7 is now rewritten and included in the new version of the manuscript. The change is indicated with red text and shaded with yellow.
Page 4, lines 197-199, and Page 5, lines 200-202.
Gelatin zymography is a powerful and useful technique for measuring the relative amounts of active and inactive gelatinases (zymogens) in aqueous samples by measuring gelatin hydrolysis. The enzymes MMP-2 (gelatinase A) and MMP-9 (gelatinase B) are fractionated in SDS-PAGE gels, to be activated by denaturation and detected by including gelatin as a substrate in the gel. The gelatin hydrolysis produce two or three white bands, for each, zymogen or active forms, after staining with Coomassie blue [12].
Reply to the Review Report (Reviewer 2)
Journal Title: Biomolecules
Article Title: ´´Mesenchymal–Epithelial Transition in Fibroblasts of Human Normal Lungs and Interstitial Lung Diseases ´´.
Thanks a lot to editors and reviewers for commenting on our manuscript.
Following the reviewer suggestions, we made several changes to the manuscript. Changes are written in red and shaded with yellow.
Below is our point-by-point response to the reviewer's comments and the actions taken to amend the manuscript, indicating the page where the change appears, as well as the new references as they appear in the new manuscript. We hope that the revised manuscript is acceptable to be published in the prominent journal Biomolecules.
Author's Reply to the Review Report (Reviewer 2)
Reviewer’s comments:
Open Review
(x) I would not like to sign my review report
( ) I would like to sign my review report
English language and style
( ) Extensive editing of English language and style required
( ) Moderate English changes required
( ) English language and style are fine/minor spell check required
(x) I don't feel qualified to judge about the English language and style
Yes Can be improved Must be improved Not applicable
Does the introduction provide sufficient background and include all relevant references?
(x) ( ) ( ) ( )
Is the research design appropriate?
(x) ( ) ( ) ( )
Are the methods adequately described?
(x) ( ) ( ) ( )
Are the results clearly presented?
(x) ( ) ( ) ( )
Are the conclusions supported by the results?
(x) ( ) ( ) ( )
Comments and Suggestions for Authors
The authors have addressed the experimental questions I raised in the first review. I am happy for the manuscript to be published after a few minor changes in the text
I believe the manuscript would benefit from the authors expanding the text in results section 3.6. The authors jump straight into there MVs work without introducing what MVs are and what are there functions. In addition it would help if the authors explained their rationale for looking at the role of MVs at the beginning of this section.
The authors should carefully review the new text in section 2.7. There are a few grammatical errors in this
new section
Submission Date
31 January 2021
Date of this review
16 Feb 2021 13:18:32
Reply to the Review Report (Reviewer 2)
I believe the manuscript would benefit from the authors expanding the text in results section 3.6. The authors jump straight into there MVs work without introducing what MVs are and what are there functions. In addition it would help if the authors explained their rationale for looking at the role of MVs at the beginning of this section.
RESPONSE: A brief description of the MVs characteristics and their functions is now included in section 3.6. The change is indicated with red text and shaded with yellow. Page 15, lines 408-415.
MVs are heterogeneous, membrane-bound sacs, shed from the surface of several cell types, having a size from 100 to 1000 nm, capable of encapsulating and transferring multiple forms of cargo, including proteins, mRNA, and miRNAs, siRNA, and plasmid DNA. MVs have relevant roles in transforming the extracellular environment, intercellular signaling, facilitating cell invasion through cell-independent matrix proteolysis, and modifying genetic expression and cell physiology in the proximal or distal recipient cells. MVs are found in multiple bodily fluids and tissues and could significantly impact future diagnostic and therapeutic strategies [36].
The authors should carefully review the new text in section 2.7. There are a few grammatical errors in this new section
RESPONSE: Thanks for the observation. The gelatin zymography method described in section 2.7 is now rewritten and included in the new version of the manuscript. The change is indicated with red text and shaded with yellow.
Page 4, lines 197-199, and Page 5, lines 200-202.
Gelatin zymography is a powerful and useful technique for measuring the relative amounts of active and inactive gelatinases (zymogens) in aqueous samples by measuring gelatin hydrolysis. The enzymes MMP-2 (gelatinase A) and MMP-9 (gelatinase B) are fractionated in SDS-PAGE gels, to be activated by denaturation and detected by including gelatin as a substrate in the gel. The gelatin hydrolysis produce two or three white bands, for each, zymogen or active forms, after staining with Coomassie blue [12].
Reply to the Review Report (Reviewer 2)
Journal Title: Biomolecules
Article Title: ´´Mesenchymal–Epithelial Transition in Fibroblasts of Human Normal Lungs and Interstitial Lung Diseases ´´.
Thanks a lot to editors and reviewers for commenting on our manuscript.
Following the reviewer suggestions, we made several changes to the manuscript. Changes are written in red and shaded with yellow.
Below is our point-by-point response to the reviewer's comments and the actions taken to amend the manuscript, indicating the page where the change appears, as well as the new references as they appear in the new manuscript. We hope that the revised manuscript is acceptable to be published in the prominent journal Biomolecules.
Author's Reply to the Review Report (Reviewer 2)
Reviewer’s comments:
Open Review
(x) I would not like to sign my review report
( ) I would like to sign my review report
English language and style
( ) Extensive editing of English language and style required
( ) Moderate English changes required
( ) English language and style are fine/minor spell check required
(x) I don't feel qualified to judge about the English language and style
Yes Can be improved Must be improved Not applicable
Does the introduction provide sufficient background and include all relevant references?
(x) ( ) ( ) ( )
Is the research design appropriate?
(x) ( ) ( ) ( )
Are the methods adequately described?
(x) ( ) ( ) ( )
Are the results clearly presented?
(x) ( ) ( ) ( )
Are the conclusions supported by the results?
(x) ( ) ( ) ( )
Comments and Suggestions for Authors
The authors have addressed the experimental questions I raised in the first review. I am happy for the manuscript to be published after a few minor changes in the text
I believe the manuscript would benefit from the authors expanding the text in results section 3.6. The authors jump straight into there MVs work without introducing what MVs are and what are there functions. In addition it would help if the authors explained their rationale for looking at the role of MVs at the beginning of this section.
The authors should carefully review the new text in section 2.7. There are a few grammatical errors in this
new section
Submission Date
31 January 2021
Date of this review
16 Feb 2021 13:18:32
Reply to the Review Report (Reviewer 2)
I believe the manuscript would benefit from the authors expanding the text in results section 3.6. The authors jump straight into there MVs work without introducing what MVs are and what are there functions. In addition it would help if the authors explained their rationale for looking at the role of MVs at the beginning of this section.
RESPONSE: A brief description of the MVs characteristics and their functions is now included in section 3.6. The change is indicated with red text and shaded with yellow. Page 15, lines 408-415.
MVs are heterogeneous, membrane-bound sacs, shed from the surface of several cell types, having a size from 100 to 1000 nm, capable of encapsulating and transferring multiple forms of cargo, including proteins, mRNA, and miRNAs, siRNA, and plasmid DNA. MVs have relevant roles in transforming the extracellular environment, intercellular signaling, facilitating cell invasion through cell-independent matrix proteolysis, and modifying genetic expression and cell physiology in the proximal or distal recipient cells. MVs are found in multiple bodily fluids and tissues and could significantly impact future diagnostic and therapeutic strategies [36].
The authors should carefully review the new text in section 2.7. There are a few grammatical errors in this new section
RESPONSE: Thanks for the observation. The gelatin zymography method described in section 2.7 is now rewritten and included in the new version of the manuscript. The change is indicated with red text and shaded with yellow.
Page 4, lines 197-199, and Page 5, lines 200-202.
Gelatin zymography is a powerful and useful technique for measuring the relative amounts of active and inactive gelatinases (zymogens) in aqueous samples by measuring gelatin hydrolysis. The enzymes MMP-2 (gelatinase A) and MMP-9 (gelatinase B) are fractionated in SDS-PAGE gels, to be activated by denaturation and detected by including gelatin as a substrate in the gel. The gelatin hydrolysis produce two or three white bands, for each, zymogen or active forms, after staining with Coomassie blue [12].
Reply to the Review Report (Reviewer 2)
Journal Title: Biomolecules
Article Title: ´´Mesenchymal–Epithelial Transition in Fibroblasts of Human Normal Lungs and Interstitial Lung Diseases ´´.
Thanks a lot to editors and reviewers for commenting on our manuscript.
Following the reviewer suggestions, we made several changes to the manuscript. Changes are written in red and shaded with yellow.
Below is our point-by-point response to the reviewer's comments and the actions taken to amend the manuscript, indicating the page where the change appears, as well as the new references as they appear in the new manuscript. We hope that the revised manuscript is acceptable to be published in the prominent journal Biomolecules.
Author's Reply to the Review Report (Reviewer 2)
Reviewer’s comments:
Open Review
(x) I would not like to sign my review report
( ) I would like to sign my review report
English language and style
( ) Extensive editing of English language and style required
( ) Moderate English changes required
( ) English language and style are fine/minor spell check required
(x) I don't feel qualified to judge about the English language and style
Yes Can be improved Must be improved Not applicable
Does the introduction provide sufficient background and include all relevant references?
(x) ( ) ( ) ( )
Is the research design appropriate?
(x) ( ) ( ) ( )
Are the methods adequately described?
(x) ( ) ( ) ( )
Are the results clearly presented?
(x) ( ) ( ) ( )
Are the conclusions supported by the results?
(x) ( ) ( ) ( )
Comments and Suggestions for Authors
The authors have addressed the experimental questions I raised in the first review. I am happy for the manuscript to be published after a few minor changes in the text
I believe the manuscript would benefit from the authors expanding the text in results section 3.6. The authors jump straight into there MVs work without introducing what MVs are and what are there functions. In addition it would help if the authors explained their rationale for looking at the role of MVs at the beginning of this section.
The authors should carefully review the new text in section 2.7. There are a few grammatical errors in this
new section
Submission Date
31 January 2021
Date of this review
16 Feb 2021 13:18:32
Reply to the Review Report (Reviewer 2)
I believe the manuscript would benefit from the authors expanding the text in results section 3.6. The authors jump straight into there MVs work without introducing what MVs are and what are there functions. In addition it would help if the authors explained their rationale for looking at the role of MVs at the beginning of this section.
RESPONSE: A brief description of the MVs characteristics and their functions is now included in section 3.6. The change is indicated with red text and shaded with yellow. Page 15, lines 408-415.
MVs are heterogeneous, membrane-bound sacs, shed from the surface of several cell types, having a size from 100 to 1000 nm, capable of encapsulating and transferring multiple forms of cargo, including proteins, mRNA, and miRNAs, siRNA, and plasmid DNA. MVs have relevant roles in transforming the extracellular environment, intercellular signaling, facilitating cell invasion through cell-independent matrix proteolysis, and modifying genetic expression and cell physiology in the proximal or distal recipient cells. MVs are found in multiple bodily fluids and tissues and could significantly impact future diagnostic and therapeutic strategies [36].
The authors should carefully review the new text in section 2.7. There are a few grammatical errors in this new section
RESPONSE: Thanks for the observation. The gelatin zymography method described in section 2.7 is now rewritten and included in the new version of the manuscript. The change is indicated with red text and shaded with yellow.
Page 4, lines 197-199, and Page 5, lines 200-202.
Gelatin zymography is a powerful and useful technique for measuring the relative amounts of active and inactive gelatinases (zymogens) in aqueous samples by measuring gelatin hydrolysis. The enzymes MMP-2 (gelatinase A) and MMP-9 (gelatinase B) are fractionated in SDS-PAGE gels, to be activated by denaturation and detected by including gelatin as a substrate in the gel. The gelatin hydrolysis produce two or three white bands, for each, zymogen or active forms, after staining with Coomassie blue [12].
Reply to the Review Report (Reviewer 2)
Journal Title: Biomolecules
Article Title: ´´Mesenchymal–Epithelial Transition in Fibroblasts of Human Normal Lungs and Interstitial Lung Diseases ´´.
Thanks a lot to editors and reviewers for commenting on our manuscript.
Following the reviewer suggestions, we made several changes to the manuscript. Changes are written in red and shaded with yellow.
Below is our point-by-point response to the reviewer's comments and the actions taken to amend the manuscript, indicating the page where the change appears, as well as the new references as they appear in the new manuscript. We hope that the revised manuscript is acceptable to be published in the prominent journal Biomolecules.
Author's Reply to the Review Report (Reviewer 2)
Reviewer’s comments:
Open Review
(x) I would not like to sign my review report
( ) I would like to sign my review report
English language and style
( ) Extensive editing of English language and style required
( ) Moderate English changes required
( ) English language and style are fine/minor spell check required
(x) I don't feel qualified to judge about the English language and style
Yes Can be improved Must be improved Not applicable
Does the introduction provide sufficient background and include all relevant references?
(x) ( ) ( ) ( )
Is the research design appropriate?
(x) ( ) ( ) ( )
Are the methods adequately described?
(x) ( ) ( ) ( )
Are the results clearly presented?
(x) ( ) ( ) ( )
Are the conclusions supported by the results?
(x) ( ) ( ) ( )
Comments and Suggestions for Authors
The authors have addressed the experimental questions I raised in the first review. I am happy for the manuscript to be published after a few minor changes in the text
I believe the manuscript would benefit from the authors expanding the text in results section 3.6. The authors jump straight into there MVs work without introducing what MVs are and what are there functions. In addition it would help if the authors explained their rationale for looking at the role of MVs at the beginning of this section.
The authors should carefully review the new text in section 2.7. There are a few grammatical errors in this
new section
Submission Date
31 January 2021
Date of this review
16 Feb 2021 13:18:32
Reply to the Review Report (Reviewer 2)
I believe the manuscript would benefit from the authors expanding the text in results section 3.6. The authors jump straight into there MVs work without introducing what MVs are and what are there functions. In addition it would help if the authors explained their rationale for looking at the role of MVs at the beginning of this section.
RESPONSE: A brief description of the MVs characteristics and their functions is now included in section 3.6. The change is indicated with red text and shaded with yellow. Page 15, lines 408-415.
MVs are heterogeneous, membrane-bound sacs, shed from the surface of several cell types, having a size from 100 to 1000 nm, capable of encapsulating and transferring multiple forms of cargo, including proteins, mRNA, and miRNAs, siRNA, and plasmid DNA. MVs have relevant roles in transforming the extracellular environment, intercellular signaling, facilitating cell invasion through cell-independent matrix proteolysis, and modifying genetic expression and cell physiology in the proximal or distal recipient cells. MVs are found in multiple bodily fluids and tissues and could significantly impact future diagnostic and therapeutic strategies [36].
The authors should carefully review the new text in section 2.7. There are a few grammatical errors in this new section
RESPONSE: Thanks for the observation. The gelatin zymography method described in section 2.7 is now rewritten and included in the new version of the manuscript. The change is indicated with red text and shaded with yellow.
Page 4, lines 197-199, and Page 5, lines 200-202.
Gelatin zymography is a powerful and useful technique for measuring the relative amounts of active and inactive gelatinases (zymogens) in aqueous samples by measuring gelatin hydrolysis. The enzymes MMP-2 (gelatinase A) and MMP-9 (gelatinase B) are fractionated in SDS-PAGE gels, to be activated by denaturation and detected by including gelatin as a substrate in the gel. The gelatin hydrolysis produce two or three white bands, for each, zymogen or active forms, after staining with Coomassie blue [12].
Reply to the Review Report (Reviewer 2)
Journal Title: Biomolecules
Article Title: ´´Mesenchymal–Epithelial Transition in Fibroblasts of Human Normal Lungs and Interstitial Lung Diseases ´´.
Thanks a lot to editors and reviewers for commenting on our manuscript.
Following the reviewer suggestions, we made several changes to the manuscript. Changes are written in red and shaded with yellow.
Below is our point-by-point response to the reviewer's comments and the actions taken to amend the manuscript, indicating the page where the change appears, as well as the new references as they appear in the new manuscript. We hope that the revised manuscript is acceptable to be published in the prominent journal Biomolecules.
Author's Reply to the Review Report (Reviewer 2)
Reviewer’s comments:
Open Review
(x) I would not like to sign my review report
( ) I would like to sign my review report
English language and style
( ) Extensive editing of English language and style required
( ) Moderate English changes required
( ) English language and style are fine/minor spell check required
(x) I don't feel qualified to judge about the English language and style
Yes Can be improved Must be improved Not applicable
Does the introduction provide sufficient background and include all relevant references?
(x) ( ) ( ) ( )
Is the research design appropriate?
(x) ( ) ( ) ( )
Are the methods adequately described?
(x) ( ) ( ) ( )
Are the results clearly presented?
(x) ( ) ( ) ( )
Are the conclusions supported by the results?
(x) ( ) ( ) ( )
Comments and Suggestions for Authors
The authors have addressed the experimental questions I raised in the first review. I am happy for the manuscript to be published after a few minor changes in the text
I believe the manuscript would benefit from the authors expanding the text in results section 3.6. The authors jump straight into there MVs work without introducing what MVs are and what are there functions. In addition it would help if the authors explained their rationale for looking at the role of MVs at the beginning of this section.
The authors should carefully review the new text in section 2.7. There are a few grammatical errors in this
new section
Submission Date
31 January 2021
Date of this review
16 Feb 2021 13:18:32
Reply to the Review Report (Reviewer 2)
I believe the manuscript would benefit from the authors expanding the text in results section 3.6. The authors jump straight into there MVs work without introducing what MVs are and what are there functions. In addition it would help if the authors explained their rationale for looking at the role of MVs at the beginning of this section.
RESPONSE: A brief description of the MVs characteristics and their functions is now included in section 3.6. The change is indicated with red text and shaded with yellow. Page 15, lines 408-415.
MVs are heterogeneous, membrane-bound sacs, shed from the surface of several cell types, having a size from 100 to 1000 nm, capable of encapsulating and transferring multiple forms of cargo, including proteins, mRNA, and miRNAs, siRNA, and plasmid DNA. MVs have relevant roles in transforming the extracellular environment, intercellular signaling, facilitating cell invasion through cell-independent matrix proteolysis, and modifying genetic expression and cell physiology in the proximal or distal recipient cells. MVs are found in multiple bodily fluids and tissues and could significantly impact future diagnostic and therapeutic strategies [36].
The authors should carefully review the new text in section 2.7. There are a few grammatical errors in this new section
RESPONSE: Thanks for the observation. The gelatin zymography method described in section 2.7 is now rewritten and included in the new version of the manuscript. The change is indicated with red text and shaded with yellow.
Page 4, lines 197-199, and Page 5, lines 200-202.
Gelatin zymography is a powerful and useful technique for measuring the relative amounts of active and inactive gelatinases (zymogens) in aqueous samples by measuring gelatin hydrolysis. The enzymes MMP-2 (gelatinase A) and MMP-9 (gelatinase B) are fractionated in SDS-PAGE gels, to be activated by denaturation and detected by including gelatin as a substrate in the gel. The gelatin hydrolysis produce two or three white bands, for each, zymogen or active forms, after staining with Coomassie blue [12].
